



# Simulations of winter ozone in the Upper Green River Basin, Wyoming, using WRF-Chem

Shreta Ghimire[1], Zachary J. Lebo[1], Shane Murphy[1], Stefan Rahimi[2], and Trang Tran[3]

[1]Department of Atmospheric Science, University of Wyoming
[2]Institute of Environment and Sustainability, University of California Los Angeles
[3]Desert Research Institute

**Correspondence:** Zachary J. Lebo (zlebo@uwyo.edu)

**Abstract.**

In both the Upper Green River Basin (UGRB) of Wyoming and the Uintah Basin of Utah, strong wintertime ozone ($O_3$) formation episodes leading to $O_3$ concentrations exceeding the 8-hour $O_3$ NAAQS (70 ppb) have been observed over the last two decades. Wintertime $O_3$ events in the UGRB were first observed in 2005 and since then have continued to be observed in-
termittently when meteorological conditions are favorable, despite significant efforts to reduce emissions. While $O_3$ formation has been successfully simulated using observed volatile organic compound (VOC) and nitrogen oxide ($NO_X$) concentrations, successful simulation of these wintertime episodes using emission inventories in a 3-D photochemical model has remained elusive. An accurate 3-D photochemical model driven by an emission inventory is critical to understand which emission sources have the most impact on $O_3$ formation. In the winter of 2016-2017 (December 2016 - March 2017) several high $O_3$ events
were recorded with concentrations exceeding 70 ppb. This study uses the Weather Research Forecasting model with chemistry (WRF-Chem) to simulate one of the high $O_3$ events observed in the UGRB during March of 2017. The WRF-Chem simulations were carried out using the 2014 edition of the Environmental Protection Agency National Emissions Inventory (EPA-NEI 2014v2), which includes estimates of emissions from non-point oil and gas production sources. Simulations were carried out with two different chemical mechanisms: the Model for Ozone and Related Chemical Tracers (MOZART) and the Regional
Atmospheric Chemistry Mechanism (RACM), and the results were compared with the observed data from 7 weather and air quality monitoring stations in the UGRB operated by Wyoming Department of Environmental Quality (WYDEQ). The simulated meteorology compared favorably to observations in terms of predicting temperature inversions and surface temperature and wind speeds. Notably, because of snow cover present in the basin, the photolysis surface albedo was modified in all simulations. Without this modification, none of the simulations formed $O_3$ exceeding 70 ppb, though the models were relatively
insensitive to the exact photolysis albedo if it was over 0.65. The MOZART simulation produced more $O_3$ in the basin than the RACM simulation and compares better with the observations. However, while $O_3$ precursors $NO_X$ and NMHC are predicted similarly in simulations with both chemistry mechanisms, simulated NMHC mixing ratios are a factor of six lower than the observations, while $NO_X$ mixing ratios are also underpredicted but are much closer to the observations within the region of oil and gas production. The results show that both the RACM and MOZART chemical mechanisms were able to produce $O_3$ even
though the NMHC mixing ratios in the model were a factor of six too low, an intriguing result for future studies.



# 1 Introduction

Tropospheric ozone ($O_3$) is a secondary pollutant harmful to human health, plants, and other animals (Fuhrer et al., 1997; Ebi and McGregor, 2008) when at elevated levels. The current 2015 US National Ambient Air Quality Standard (NAAQS) for the 8-h average $O_3$ mixing ratio is 70 parts per billions (ppb) (EPA)[1]. As of August 14, 2020, the 2015 NAAQS standard for the 8-h average $O_3$ mixing ratio has been proposed to be retained (EPA, 2020). Any hourly occurrence of $O_3$ concentration greater or equal to the NAAQS standard is referred to as an $O_3$ event throughout this paper. In the past decades, there has been a significant increase in wintertime as well as summertime $O_3$ events in the western US (Cooper et al., 2012).

According to the US Energy Information Administration (EIA), in 2018, Wyoming was the 8th largest producer of oil and natural gas in the United States, with a majority of the natural gas production coming from the Upper Green River Basin (UGRB). Specifically, the UGRB accounts for 60% of the state's natural gas production and 16% of its oil production (Wyoming State Geological Survey; WSGS, 2020). As of 2017, there were 5506 total wells (5436 producing wells) in the Jonah and Pinedale fields that constitute the UGRB, a 5.7% increase in the total and 5.9% increase in the producing wells in the UGRB compared with those in 2016 (http://pipeline.wyo.gov/FieldReportYear.cfm). By September 2020, there had been 8.8% of increase in the total wells since 2017 and 14.6% increase in producing wells in the UGRB.

The formation of $O_3$ has traditionally been an urban summertime phenomena because of the need for strong solar intensity and sufficient Volatile Organic Compound (VOC). Elevated concentrations of wintertime $O_3$ in a few rural US basins have been associated with the rapid development of natural gas and oil production fields (Mansfield and Hall, 2013; Edwards et al., 2014; Ahmadov et al., 2015; Field et al., 2015a, b). Such elevated $O_3$ events can occur in winter under specific meteorological conditions: a snow-covered ground that provides high albedo that increases solar intensity while also preventing solar heating of the ground (Carter and Seinfeld, 2012) and weak/calm winds. Combined, these conditions result in a persistent temperature inversion and little horizontal/vertical transport, which provides the conditions needed for the photochemical production and build up of $O_3$ (Mansfield and Hall, 2018).

Several studies have been carried out to understand the meteorological and chemical processes leading to high wintertime $O_3$ events in western US oil and gas basins. These studies have focused on observational measurements (Schnell et al., 2009; Oltmans et al., 2014b; Rappenglück et al., 2014; Field et al., 2015b; Lyman and Tran, 2015), aircraft measurements (Oltmans et al., 2014a), statistical models (Mansfield and Hall, 2013), box models (Carter and Seinfeld, 2012; Edwards et al., 2013, 2014), and 3-D photochemical models (Rodriguez et al., 2009; Ahmadov et al., 2015). Most of these studies have been carried out in the UGRB and Utah's Uintah Basin (UB) and both basins have been identified as regions exceeding the NAAQS (Lyman and Tran, 2015). These studies have shown the principal role played by emissions from oil and natural gas production fields in the formation of wintertime $O_3$. However, the assessment of wintertime $O_3$ formation in these regions poses serious challenges because each basin has complex topography and meteorological conditions along with poorly constrained precursor; VOC and nitrogen oxide ($NO_X$) emissions. One shortfall of all previous studies is that most of them have not utilized an existing emission inventory to model $O_3$ formation. Rather, these studies have utilized observed atmospheric concentrations of precursors to

---

[1] https://www.epa.gov/criteria-air-pollutants/naaqs-table





model $O_3$ formation, thus making it difficult to assess how future expansion of production or various emission reductions will affect $O_3$ formation.

Schnell et al. (2009) summarized the confluence of three major factors for wintertime $O_3$ formation: (i) the extensive production of oil and natural gas that releases $NO_X$ and VOCs or hydrocarbons (HCs) into the atmosphere, (ii) calm wind conditions, and (iii) high albedo caused by snow accumulation at the surface that leads to a strong temperature inversion. A strong in-

version traps $O_3$ and its precursors near the ground; if the inversion persists for several days, the concentrations of $O_3$ and its precursors increase. The high surface albedo also provides additional shortwave radiation for photochemistry compared to a dry landscape.

Some studies have specifically pointed out the importance of deep snow cover or high surface albedo in the formation of wintertime $O_3$. Oltmans et al. (2014b) and Rappenglück et al. (2014) noted that in March 2011, the UGRB experienced high

hourly $O_3$ concentrations exceeding 150 ppb, which was associated with the deepest snow cover of the season. In addition, Oltmans et al. (2014b) also pointed out that for the period with snow coverage on the ground, the sum of incoming and reflected ultraviolet levels were almost 80% higher than the period with no snow cover, addressing the impact of fresh snow accumulation during high $O_3$ events. Rappenglück et al. (2014) noted a significant increase in the background $O_3$ concentration from around 40 ppb in January to 60 ppb in March 2011, owing to the changes in the meteorological and chemical processes

each month that change the pollutant concentration.

Numerous measurement studies have pointed out the important roles played by topography and both meteorological and chemical processes in the basin, leading to different $O_3$ and precursor concentrations within each basin and from year to year. Field et al. (2015b) carried out air quality measurements in the UGRB for two consecutive winters (2011 and 2012) at a site located 5 km southeast of a Wyoming Department of Environmental Quality (WYDEQ) air quality and weather monitoring

station (Boulder). They measured $O_3$, reactive nitrogen compounds, methane ($CH_4$), total non-methane hydrocarbon (NMHC), carbon monoxide (CO), and other standard meteorological parameters. The lower concentration of observed $O_3$ in 2012 were associated with lower NMHC concentrations, which was lower compared to 2011. Furthermore, Lyman and Tran (2015) measured $O_3$ and meteorological parameters at different location in the UB and observed a negative correlation between the $O_3$ concentration and station elevation. The stations at higher elevations showed very few $O_3$ exceedance events compared to

those at lower elevation. As mentioned by Schnell et al. (2009) the prolonged inversion period traps $O_3$ near the basin floor due to low wind speeds and limited vertical transport, hence reducing $O_3$ concentrations at the higher elevations. Oltmans et al. (2014a) conducted 7 aircraft flights in the UB and found that the high $O_3$ concentrations were confined in the shallow inversion layer, namely 300-400 m above the ground.

Mansfield and Hall (2013) used a statistical model to accurately predict $O_3$ formation, but they note challenges in extending

the findings from one basin to another, as factors such as thermal inversion and snow cover that play an important role in wintertime $O_3$ formation vary among basins. They used quadratic regression models to predict the daily $O_3$ concentrations in the UB and UGRB. They found that the high $O_3$ events in the UB and UGRB occurred in February and March, respectively. However, the most intense inversion periods in both basins occurred in January. For both the UB and UGRB, they concluded that these high $O_3$ events were highly sensitive to the solar radiation, which intensifies as the year progresses.





Carter and Seinfeld (2012) used a box model to study $NO_X$-limited and VOC-limited regimes in the UGRB. They found that the concentrations of NO, $NO_2$ and NMHC, and VOC/$NO_X$ ratios varied both spatially and temporally within the basin. Hence, they suggested that equal attention needs to be given to the geographical distribution of $O_3$ precursors and the local meteorology. Edwards et al. (2013) utilized the Dynamically Simple Model of Atmospheric Chemical Complexity (DSMACC), a photochemical box model with a very thorough chemical mechanism, to assess the sensitivity of $NO_X$ and VOC along with

radical precursors[2] for $O_3$ production in the UB. Using this model, with input of observed $O_3$ precursors, they were able to accurately simulate relatively small amounts of $O_3$ formation in the absence of snow cover in 2013. Edwards et al. (2014) demonstrated that the same model could simulate large amounts of $O_3$ production in the UB when snow cover was present, and they emphasized the importance of carbonyl photolysis in the radical chemistry.

      There have been a few studies that have utilized 3-D photochemical models to simulate high $O_3$ events in western US oil

and gas basins, though to date there has not been a successful 3-D photochemical modelling study that has simulated high wintertime $O_3$ in the UGRB. Rodriguez et al. (2009) applied the Comprehensive Air Quality Model with Extensions (CAMx) to assess the impacts of the development of oil and gas fields in the western US on the air quality of various parks and national wilderness areas in the inter-mountain west for 2002. They concluded that the model captured the general trend in $O_3$ on a monthly scale; however, the model did not capture wintertime $O_3$ formation events occurring during strong inversions.

Ahmadov et al. (2015) used the Weather Research Forecasting model coupled with Chemistry (WRF-Chem, version 3.5.1) to study wintertime $O_3$ pollution in the UB. To account for the emissions from the oil and gas sector, they employed two different emission scenarios. The first emission dataset was the US EPA National Emission Inventory 2011 version 1 (NEI2011; bottom-up) and the second emission dataset was derived from in situ aircraft and ground-based measurements (top-down). They reported an underestimation of hydrocarbons ($CH_4$ and other VOCs) and an overestimation of $NO_X$ emissions in the NEI2011

inventory compared to the top-down emission scenario. Ahmadov et al. (2015) found that the model simulation using the bottom-up NEI2011 inventory underestimated the high $O_3$ concentrations observed in the UB and that it was necessary to utilize observed concentrations of VOCs and $NO_X$ to successfully simulate observed $O_3$ mixing ratios.

      As outlined above, wintertime $O_3$ production requires a thermal inversion as well as sufficiently deep snow (i.e., deep enough to cover most of the vegetation) over a larger area; hence, not all winters experience high $O_3$ concentrations. Additionally,

reported emissions from oil and gas have been significantly reduced over the last decade WYDEQ (2018). In the winter of 2005, the newly installed WYDEQ monitoring stations at Boulder, Daniel, and Jonah observed multiple occurrences of high $O_3$ concentrations that exceeded the existing 1997 8-hour $O_3$ standard (84 ppb, WYDEQ, 2018). Since 2005, WYDEQ has operated regular annual $O_3$ monitoring in the UGRB, and several air quality and weather monitoring stations have been added in the basin. In recent years (most notably 2008, 2011, 2017, 2019 and 2020), elevated wintertime $O_3$ events have been observed

in the UGRB, with hourly $O_3$ concentrations exceeding 70 ppb for several days in each year. The formation and occurrence of elevated wintertime $O_3$ concentrations is an unusual event compared to its urban summertime formation. In July 2012, the UGRB was declared as a marginal non-attainment area for $O_3$ by the US EPA (Rappenglück et al., 2014). In the winter of 2012, there were only 3 days in which the 8-hour averaged $O_3$ mixing ratios exceeded 75 ppb (NAAQS 2008), while in the

---

[2]Formaldehyde, nitrous acid and nitryl chloride





winter of 2011, there were 7 days of exceedance (Field et al., 2015b) at a site located near the Boulder station. In March 2017,
the Boulder station observed several hours of an hourly averaged $O_3$ concentration exceeding 70 ppb (NAAQS 2015).

Given the continued occurrence of high $O_3$ events in the UGRB, the lack of modeling studies aimed at understanding the
formation of $O_3$, and plans to continue development of the basin, it is important to develop a photochemical model capable
of reproducing high $O_3$ events of the recent past in order to understand how events can be prevented in the future. The main
goal in this study is to assess if a photochemical model (particularly WRF-Chem) operating with NEI emissions can simulate
wintertime $O_3$ formation in the UGRB. Successful simulation of $O_3$ events would mean the model could then be utilized to
assess effective emission control in preventing future $O_3$ events as well as the impact of future development on $O_3$ formation.
This study primarily focuses on one of the elevated wintertime $O_3$ events in the winter of 2017; a 4-day period from Mar 3
to Mar 7, 2017, because 2017 was an active year for elevated $O_3$ in the UGRB (WYDEQ, 2018). The observed hourly $O_3$
mixing ratios during the period exceeded 70 ppb (NAAQS 2015) for several hours at several air quality monitoring stations in
the UGRB. For our $O_3$ simulations, we have chosen to simulate the 2017 season because this was the most recent year with
sustained periods of high $O_3$ when this project began in 2019. It is most useful to simulate $O_3$ events from recent years (versus
modeling events in 2011) because basin-wide emission estimates from the State DEQ have decreased significantly over the last
decade. Successful simulation of an $O_3$ event in 2011 would not be terribly meaningful for assessment of the model's ability
to simulate $O_3$ formation under current emission levels given emission levels and VOC:NOx ratios are estimated to have been
significantly different in 2011, the only year in which vertical data are available (see below). In this paper, the results from
WRF-Chem simulations for the given period are analyzed, aimed at understanding the production of $O_3$ in the UGRB.

## 2 Methods

This section describes the study area, model setup, datasets, methods, and preprocessing tools utilized in the WRF-Chem
simulations and to validate the model results.

### 2.1 Study Region

The focus area of this study is the UGRB. The UGRB is a valley located in Sublette County in western Wyoming, with the
Wyoming Range to its west, the Gros Ventre Range to its north, and the Wind River Range to its east. There are 7 weather
and air quality monitoring stations operated by the WYDEQ in or near the UGRB: BP - Big Piney, B - Boulder, DS - Daniel
South, JS - Juel Spring, M - Moxa Arch, P - Pinedale and SP - South Pass, whose exact locations are shown in the upper panel
of Figure 1. In addition, the geographical information related to these stations is provided in Table 1. Five of the stations (BP,
B, DS, JS, and P) are in close proximity to each other and lie in the basin where wind and pollutant transport can be affected
by the mountains to the east, west, and north. Stations B and P lie in close proximity to the Pinedale Anticline and Jonah Field
Developments (PAJF). The natural gas and oil development fields are located southwest of stations B and P, as shown in the
bottom panel of Figure 1 (Toner et al., 2019). The other two stations (M and SP) lie further away from the basin. Station SP is





located in the foothills of the Wind River Range and has the highest elevation, and station M is the southernmost and lowest in elevation and is located in close proximity to an interstate highway (I-80).

## 2.2    Model Setup

Simulations of $O_3$ formation in the UGRB were conducted using WRF-Chem (Skamarock et al., 2008) version 3.9.1. WRF-Chem is a fully coupled model, in which its atmospheric chemistry component is directly coupled to the meteorological

component of the model (Grell et al., 2005). The meteorological and air quality components of the model use the same transport and physics schemes as well as the same vertical and horizontal grid structure. The baseline model configuration with the physical parameterizations used for the study is shown in Table 2. Figure 1 shows the model domain and terrain height, which is centered on the UGRB. The model domain is represented by a grid of 200 x 200 x 60 points with a horizontal grid spacing of 4 km; vertical grids extend up to 100 hPa, with 60-m grid spacing near the surface and 250-m grid spacing at the top of the

model.

## 2.3    Datasets

The National Centers for Environmental Prediction (NCEP) North American Regional Reanalysis (NARR) (Mesinger et al., 2006) was used for the initial and boundary meteorological conditions for the simulations in this study. The data are available on a Lambert conformal conical grid with a grid spacing of approximately 0.3 degrees (32 km). The 3-hourly fields with 29

vertical pressure levels from 1000 to 100 hPa were used in this study to initialize and provide the lateral boundary conditions for the WRF-Chem simulations to study $O_3$ formation.

The NEI data were used for emissions in the the WRF-Chem simulations. The data for natural gas and oil sources were obtained from the US EPA NEI-2014 dataset (version 2, hereafter; NEI2014v2) released in February 2018 (US-EPA, 2018). The NEI2014v2 data were the latest emission inventory available at the time of the initiation of this study and is available

at a 12-km horizontal resolution. This particular version of the emission dataset incorporates the processes associated with the exploration, drilling, and production of oil, gas, and coal-bed $CH_4$ wells in the UGRB. The EPA emission estimates are the most widely used and easily available estimate that include most potential emission sources that could impact air quality. However, previous comparisons by Alvarez et al. (2018); Robertson et al. (2020) have pointed out underestimations of $CH_4$ emissions for the oil and gas extraction basins in EPA estimates compared to their observations.

The observed meteorological and air quality data from the aforementioned 7 weather and air quality monitoring stations were obtained from the WYDEQ website. The data are available in 5-minute and hourly formats. The hourly data were used for this study for a direct comparison of meteorological parameters, such as temperature and wind speed, and chemical species, such as $O_3$, $NO_X$, $CH_4$, and NMHC, with the simulated results. The NHMC data were only available at the Boulder station as this was the only station equipped to report these results.



## 2.4 Preprocessing

The EPA anthro emiss tool provided by the Atmospheric Chemistry Observations & Modeling (ACOM) division at the National Center for Atmospheric Research (NCAR) was used for preprocessing the emissions in this study. This tool creates anthropogenic emission files from the NEI datasets for lat/lon grids that can be ingested into the WRF model. The MOZART and RACM chemistry mechanisms use different species grouping; hence, the emission inventory files were processed separately for each mechanism. Mozbc, which is also provided by ACOM, was also used in this study. The mozbc tool maps the species from the Community Atmosphere Model with Chemistry (CAM-Chem) global dataset to WRF fields that can easily be ingested into WRF-Chem as initial and boundary conditions.

For MOZ17, two other WRF-Chem utilities were also used: exo_coldens and wesely. The exo_coldens utility helps read $O_3$ and $O_2$ climatological atmospheric column values rather than using fixed values, and this is coupled to an updated photolysis option (photo_opt=4). For dry deposition in MOZART, an additional file is required that allows for seasonal changes in dry deposition. The additional information is provided using the wesely utility. Both the exo_coldens and wesely utilities read the WRF input files as well as emission files for the MOZART chemistry mechanism to produce additional data files that can be read by the WRF-Chem model.

The NEI2014v2 dataset provides emissions covering the model domain, but the advection of chemical species into the domain through the lateral boundaries must also be considered. The WRF-Chem simulations in this study used the NEI2014v2 emission data re-gridded to the WRF-Chem domain. The initial and boundary conditions of the simulations were updated every 24 hours for each simulations using the CAM-Chem data (Emmons et al., 2020).

## 2.5 WRF-Chem simulations

The $O_3$ formation simulations focus on a 4-day period from Mar 3 to Mar 7, 2017. For all simulations, the model physics and photolysis surface albedos were modified to account for the effect of snow on photolysis in the model. The default photolysis albedo in the model is 0.15 because the model was primarily developed for summertime photochemistry. The default photolysis albedo is much lower than what is commonly observed during winter when the surface is covered with snow. Under the default albedo of 0.15, the simulations drastically underestimated $O_3$ formation (as shown in the results below). This study is intended to study *wintertime* photochemistry of $O_3$. We thus require a higher albedo to represent a snow-covered surface. Hence, in an effort to simulate a range of potential surface conditions, multiple albedo sensitivity simulations were carried out. A similar study using WRF-Chem with RACM chemistry was carried out by Ahmadov et al. (2015) in the UB, Utah, where they set the surface albedo to 0.85 in their simulations of wintertime $O_3$ production. As noted by Mansfield and Hall (2013), for the wintertime $O_3$ formation the factors such as thermal inversion and snow cover play an important role and they vary among the basins. Hence the findings and characteristics of wintertime $O_3$ formation cannot be extended from one basin to another. Specially, surface albedos of 0.55, 0.65, 0.75, 0.85 and 0.95 were used for the sensitivity study and fixed to 0.85 in the model for further analysis based previous estimates of snow albedo in the region (Ahmadov et al., 2015) and sufficient $O_3$ formation in the UGRB using 0.85 surface albedo.





In this study, two different chemistry mechanisms are used: (i) the Model for Ozone and Related Tracers (MOZART) and (ii) the Regional Atmospheric Chemistry Mechanism (RACM). The MOZART chemistry mechanism has been widely used model to study $O_3$ formation and transport around the world (Hauglustaine et al., 1998; Murazaki and Hess, 2006; Beig and Singh, 2007; Yarragunta et al., 2019). In the UB, RACM has been successfully used to simulate $O_3$ production due to oil and natural gas production in winter when observed levels of VOCs and $NO_X$ were inputs (Ahmadov et al., 2015). Based on the findings from Ahmadov et al. (2015), the important point noted by Mansfield and Hall (2013), and the MOZART and RACM mechanisms being widely used chemical mechanisms to study $O_3$ both globally and regionally, the simulations were carried out with these two chemical mechanisms to understand which chemical mechanism provided the best comparison with observed $O_3$ and its precursors in the UGRB. Hereafter, the simulation using MOZART chemistry will be referred to as MOZ17 and that with RACM chemistry as RACM17 based on the chemistry mechanism used and the year of the study period. The WRF-Chem namelist options used for MOZ17 and RACM17 are provided in the supplemental section A2 of this paper in Figures A2 and A3, respectively. Additionally, some key points that were considered to achieve the goals of this study and needed to reproduce the results are as follows: (i) sufficient surface albedo to represent the effect of snow cover and depth on the meteorological conditions, (ii) correct photolysis albedo to represent the wintertime conditions for the chemical mechanisms to reproduce sufficient $O_3$, and (iii) NEI data as well as CAM-CHEM global emissions data processed separately for each chemical mechanisms, as different mechanisms lump chemical species differently and are also driven by different chemical reactions..

## 2.6 Temperature Inversion Analysis and Surface Meteorology

To study the ability of the model to replicate observed meteorological conditions in the UGRB, we study the temperature inversion, weak winds, and surface temperature. The temperature inversion was studied using the WRF model (without chemistry) for 2011 with the same meteorological setup, while for weak winds and surface temperature WRF-Chem simulations for 2017 were utilized, owing to differences in data availability between the different periods. For model validation, the simulation results were compared with vertical profiles of temperature and $O_3$ from ozonesonde data collected during two intensive operational period (IOPs) in 2011. The temperature inversion was studied to validate the ability of WRF model meteorology to simulate inversions in the basin. The data from year 2011 was utilized since the WYDEQ Air Quality Department (AQD) conducted two IOPs in winter 2011 (MSI, 2011, Feb 28 to Mar 2 and Mar 9 to Mar 12). This is the only year for which vertically resolved meteorological data were available from radiosondes. The observed vertical data for the temperature inversion was also obtained from the WYDEQ website. The IOP events were identified based on the conditions (deep snow and large spatial coverage in the study area, development of an inversion, and calm surface winds) that support elevated $O_3$ concentrations. During each IOP period, 3-4 ozonesondes were launched adjacent to the Boulder station (see Fig. 1) each day, providing vertical profiles of $O_3$ mixing ratio, temperature, and wind speed. The WRF simulation was carried out for the entire winter of 2011 (Dec 1 2010 to Mar 31 2011), which includes both IOP periods and the high $O_3$ events of the winter of 2011. We understand that the ability of the model to simulate one event (i.e., the vertical structure for a few days in 2011) does not indicate that it will perform accurately again. However, with the lack of data, we are forced to either not examine the vertical structure at all





or instead find an analog that can provide some level of confidence in the model's ability to replicate the vertical structure of the lower troposphere during high-$O_3$ events. We chose the later and proceeded with the no chemistry simulations for the IOPs in 2011. The simulation will hereafter referred to as IOP11.

260 ## 3 Results and Discussion

To simulate $O_3$ formation in the UGRB, we first validated the WRF model's performance in simulating the observed vertical temperature profile and surface meteorology during strong inversions. After determining that WRF was able to reasonably reproduce the meteorological conditions necessary for $O_3$ formation, we studied $O_3$ formation with the WRF-Chem model using two different chemical mechanisms.

265 ### 3.1 Validation of WRF Model Meteorology

### 3.1.1 Temperature Inversion

Owing to the importance of thermal inversions for the build up of $0_3$ in wintertime events, we first explored the ability of the model to simulate temperature inversions within the selected modeling framework. Vertical profiles of the observed temperature and $O_3$ mixing ratio during the most recent intensive operating periods (IOP) (Feb 28 to Mar 2 and Mar 9 to Mar 12 2011) were 270 compared with the simulated vertical temperature profiles from simulations with WRF during the same time period (IOP11, Figure 2). Although 7 days were identified as the IOP period, the results from only 4 days are discussed due to ozonesonde data availability. Because these runs were completed to compare meteorology and not chemistry, the WRF model without chemistry was used and simulated $O_3$ is not available. We did not aim to simulate $O_3$ events from 2011 because emissions have changed dramatically since 2011 and there is not a good inventory that includes oil and gas sources for that period. Observed $O_3$ is 275 presented to demonstrate how $O_3$ formation follows the inversion events.

A shallow mixing height can be seen in each profile. The residual layer above the ground appears to be well mixed early in the simulation; hence, we can see fairly uniform $O_3$ concentrations in the vertical. High concentrations of $O_3$ were observed on Mar 1-2, 2011. On these days, a strong inversion is observed with a shallow mixing height of around 500 m agl, which prevents vertical mixing thus leading to a build up of $O_3$ precursors that then lead to high concentrations of $O_3$ that increase 280 in the afternoon MSI (2011). On Mar 2, 2011 (third row), higher morning $O_3$ was observed compared to the previous day, presumably due to the persistent inversion, which is validated by the observation of high hydrocarbon concentrations in the afternoon of Mar 2 (MSI, 2011).

For the days discussed here, the simulated temperature is 2 to 4 °C warmer than the observed temperature, except for Mar 9, 2011 (Figure 2, last row), where it is 2 to 5 °C colder than the observed temperature near the surface. During the morning hours, 285 the simulated temperatures follow the observed temperatures fairly well; however, the simulated inversion height is slightly elevated. In both the observations and the model, the inversion height increases through the day and the inversion strength (difference in maximum vs. surface temperature) decreases. However, the model seems to increase the inversion height slightly





too much while also decreasing the strength of the inversion. Overall the model simulation of the inversion events was deemed adequate to proceed.

### 3.1.2 Surface Meteorology

Given the model's ability to relatively accurately represent temperature inversions, at least based on our comparison with available data from 2011, we further assess the model's ability to predict surface meteorology focusing on the target period of high $O_3$ in March 2017. It is important to highlight again that vertical data are not available for the selected time period. We utilize observations from the high $O_3$ events of 2017 because the seven ground stations measure basic meteorological parameters. It is crucial for the photochemical model to simulate low temperatures and calm winds to be able to replicate high $O_3$ concentrations (Schnell et al., 2009).

The observed 2-m temperature data for Pinedale were unavailable, hence the temperature correlation for only six stations are shown in Figure 3. Both simulations show good correlation with the observed temperatures, and the correlation coefficients do not show any sensitivity to the different chemistry mechanisms at the Boulder, Moxa Arch, and South Pass stations. However, RACM17 shows higher correlation coefficients compared to MOZ17 at other stations. Although the difference in the correlation coefficients for the different chemistry mechanism is small, it is likely due to radiation feedbacks between the chemistry and meteorology in these mechanisms and internal model variability (Bassett et al., 2020). Furthermore, the temperature bias between the observed and simulated datasets is below 3 °C at all stations (Table 3), and all of the data points lie in close proximity to the one-to-one lines. Overall, the simulations show good correlation with the observed 2-m temperatures.

As mentioned earlier, calm wind speeds are an essential meteorological condition for the photochemical production of wintertime $O_3$ because they are necessary for the accumulation of $O_3$ precursors. The correlation between observed and simulated wind speeds is shown in Figure 4. The correlation coefficients are calculated for each data point (hourly) for the entire study period, although only wind speeds from 0 to 10 m s$^{-1}$ are shown given the focus of the study is calm periods. For all stations except South Pass, a majority of the data points are clustered below or around 4 m s$^{-1}$, which means that for the majority of the time, both the observed and simulated wind speeds are less than or equal to 4 m s$^{-1}$. The differences in the correlation coefficient between different simulations are due to internal model variability of the model (Bassett et al., 2020). Therefore, the relatively low correlation coefficients may be the result of small variations of low wind speeds. To test this idea and to verify that calm periods were successfully simulated when they occurred, Table 4 shows the percentage of the times the simulated and observed wind speeds are less than or equal to different thresholds (3, 4 and 5 m s$^{-1}$). For example, at Big Piney both the simulated wind speed from MOZ17 and the observed wind speed are less than or equal to 3 m s$^{-1}$ for 91.89% of the hourly periods analyzed, while for RACM17 this figure is 90.67%. Again, the chosen thresholds are based on the interest in studying calm wind speed in the basin, which enable pollutant accumulation near the surface. Therefore, even though the correlation coefficients between the modeled and observed winds are relatively low, we conclude from the results in Table 4 that WRF with either chemistry mechanism is able to successfully predict low winds the large majority of the time they occur.





### 3.2 Control Simulation and $O_3$ Production


Given the aforementioned ability of the model to accurately simulate the key meteorological conditions needed for $O_3$ production and accumulation, we now turn to the chemical mechanisms and their ability to produce the observed hourly periods with high $O_3$. At first $O_3$ formation was simulated in the UGRB using the MOZART chemistry mechanism and it was noted that the modeled concentrations were dramatically below observed $O_3$. However, the default WRF-Chem model has a low photolysis

albedo (0.15) as it was intended to simulate summertime $O_3$, which does not typically occur over high-albedo surfaces. We modified the photolysis albedo in the model based on Ahmadov et al. (2015), who noted that in the UB, it was necessary to increase the photolysis albedo to simulate $O_3$ production. In an effort to understand the sensitivity of $O_3$ formation to the photolysis albedo in the WRF-Chem model, we performed a sensitivity test. As described in the methods section, we carried out several albedo sensitivity simulations with various albedo settings ranging from 0.55 to 0.95 (spanning albedos representative

of partially snow-covered vegetation to fresh, deep snow) and compared the results to the results with the default albedo of 0.15. All of the albedo sensitivity tests used the MOZART chemical mechanism. Figure 5 compares the default albedo (0.15) with different photolysis albedo settings (0.65 and 0.85). It is evident that the default photolysis albedo produces much lower $O_3$ concentrations at all stations. However, when the model is altered to use an albedo of 0.85, the diurnal variation and high $O_3$ peaks are captured relatively well. Additionally, results from different albedo settings (0.55, 0.65, 0.75, 0.85 and 0.95) are

shown in the supplemental section of the paper (Figure B1). For the remainder of the simulations in this paper, a photolysis albedo of 0.85 is used, which is the same albedo used by Ahmadov et al. (2015) in the UB.

Setting a fixed photolysis albedo of 0.85, we next compared simulations using two different chemistry mechanisms available in WRF-Chem: MOZART and RACM. Figure 6 compares the time series of simulated hourly $O_3$ concentrations from three different simulations MOZ17, RACM17, and RACM with dry deposition turned on for all gas-phase species at the seven UGRB

monitoring stations. MOZART chemistry adjusts the dry deposition rates over snow surfaces (owing to the use of wesely preprocessing tool that adjust the season change in dry deposition), where the loss is expected to be greatly reduced. On the contrary, RACM does not adjust the dry deposition rate over such surfaces, hence the additional simulation with deposition turned off to mimic the very slow deposition of gas-phase species over a snow-covered surface (i.e., RACM17). The hourly averaged observed background daily $O_3$ mixing ratio is approximately 55 ppb at all stations. During the afternoon hours, most

of the stations have hourly $O_3$ mixing ratios greater than 70 ppb, the 8-hour NAAQS. The observed $O_3$ concentrations are highest at the Boulder site, which is likely because it lies in close proximity to the PAJF production facilities and is thus closer to the main sources of VOC precursors than the other sites. For Moxa Arch and South Pass, the observed $O_3$ concentrations are lower because they do not lie in close proximity to the wells and also lie further from the basin.

The RACM simulation with the default dry $O_3$ deposition of gas-phase species does not produce sufficient $O_3$ to replicate

the observed $O_3$ concentrations (Figure 6; purple lines). To better understand the chemistry mechanism's sensitivity to dry deposition, we compare the diurnal variation of $O_3$ concentrations from the simulation with the default dry deposition in RACM to the RACM17 simulation, where dry deposition is turned off (Figure 6; red lines) at the 7 monitoring stations. Although, the $O_3$ concentrations from the MOZ17 simulation are still higher than in RACM17, turning off the dry deposition





in RACM results in significantly higher $O_3$ concentrations than when dry deposition is allowed. The $O_3$ concentrations in

MOZ17 dissipate more slowly at night compared to RACM17. The higher concentrations of observed $O_3$ are well captured in the MOZ17 simulation. At Big Piney and Daniel South, which are located on the eastern side of the Wyoming range, both simulations overestimate the first $O_3$ event (Mar 03 2017 at 15:00 locals time). The MOZ17 simulation captures the diurnal cycle of $O_3$ reasonably well at Boulder. However, the simulations miss the higher $O_3$ concentrations at Juel Spring. Overall, both the MOZ17 and RACM17 simulations do reasonably well at simulating the $O_3$ mixing ratios in the UGRB for the selected

study period and capturing the diurnal variation of the $O_3$ concentration, a first for a photochemical model using an existing emissions inventory, although it is important to remember that this was only possible after adjusting the photolysis albedo in the model and, in the case of RACM, turning of dry deposition of gas-phase species.

       To better understand the differences in the simulated and observed $O_3$ concentrations, we next looked at the precursor ($NO_X$) concentrations. Figure 7 shows the time series of hourly $NO_X$ at the 7 monitoring stations along with results from MOZ17

and RACM17. The observed hourly mixing ratios of $NO_X$ at Big Piney, Boulder and Pinedale are higher than the other stations. These three stations are all near small towns in the region with Pinedale being the largest of the towns and Pinedale having notably higher $NO_X$ than the others. The $NO_X$ mixing ratio is primarily affected by its emission rate in the region. At Pinedale, the higher observed concentrations are most likely due to the fact that the station is near the city of Pinedale where there are sources of $NO_X$ that are not related to oil and gas, most notably residential wood burning. However, residential wood

burning is not well represented in the emission inventory; thus, the model is expected to underestimate $NO_X$ from this source. The elevated observed $NO_X$ concentrations compare well with the observed PM2.5 concentrations at Pinedale (Figure B2), which supports that wood burning is a strong $NO_X$ source in these areas. The simulated concentrations of $NO_X$ do not show any sensitivity to the different chemical mechanisms, emphasizing that the emissions dominate concentrations, not chemical loss mechanisms. The lower $NO_X$ mixing ratios are well captured at stations such as Boulder, Juel Spring and Moxa Arch

even during the high $O_3$ events. The simulations overestimate the $NO_X$ concentrations at Daniel South, although the $NO_X$ observations at this station are missing about half the time. The observed and simulated $NO_X$ concentrations at South Pass are low and show little variability, emphasizing that this station is removed from the oil and gas production region. Overall, the simulations underestimate the observed $NO_X$ concentrations to varying degrees depending on the location and do not capture the diurnal cycle well. However, the simulated $NO_X$ concentrations are relatively close to the observations at the stations

with high $O_3$ events other than at Pinedale where $NO_X$ concentrations are consistently low. The $NO_X$ concentrations are very similar between the RACM and MOZART chemical mechanisms.

       The top panel in Figure 8 compares the simulated NMHC concentrations (plotted on the left; primary $y$-axis) and observed NMHC concentrations at the Boulder station (plotted on the secondary $y$-axis). The Boulder station is the only monitoring





site in the basin that measures either NMHC or CH$_4$. In addition, the MOZART[3] and RACM[4] chemical mechanisms lump the VOC species differently. The bottom panel of Figure 8 shows the observed O$_3$ concentrations at the Boulder station during the same time period showing that the accumulation of NMHC leads to the production of O$_3$. The magnitudes of the simulated NMHC concentrations are lower by a factor of approximately 6 compared the observation. Both RACM17 and MOZ17 give very similar NMHC mixing ratios. In fact, the chemical production of O$_3$ does not remove a large amount of the NMHC present. When it was discovered that the model simulated VOC mixing ratios were dramatically different from the

observations at the Boulder site, we employed University of Wyoming mobile laboratory data to confirm that the Boulder site does not record anomalously high mixing ratios relative to the surrounding area that would all be within the same grid-cell in the model (as the station sits in a small valley). The mobile lab does not measure NMHC, but both the mobile lab and the Boulder station measure CH$_4$, enabling us to see if CH$_4$ measurements made by the lab in the region surrounding the Boulder site were significantly different than those reported by the site. Hence, we analyzed the CH$_4$ concentrations (a proxy for VOC

concentrations) collected by the mobile lab during an O$_3$ event in 2020, the closest year to our study period for which data are available. The WYDEQ Boulder site data were within 25% of the data collected by the mobile lab near the monitoring site (Figure A1). This observation indicates that the difference between simulated and observed NMHC is not the result of anomalously high mixing ratios at the Boulder site, but concluded that the NMHC mixing ratio measured at the Boulder site is an accurate representation in the region. Although the overall temporal trend in the NMHC mixing ratio is well captured by

the simulations, both MOZ17 and RACM17 dramatically underpredict the NMHC mixing ratios.

It is very intriguing that both chemical mechanisms are able to reasonably replicate the O$_3$ concentrations at the monitoring sites despite the fact that NMHC concentrations in the model are approximately 6 times lower than those observed at the Boulder monitoring site. The mobile lab results strongly suggest that this discrepancy is not due to non-representative measurements at the Boulder monitoring site. This leaves the possibilities that the simulated NMHC are much more reactive than the actual

NMHC, that some other feature of the chemistry is too active in the model, that the UGRB will continue to experience high O$_3$ events even at much lower NMHC levels, or finally that the models are extremely sensitive to the exact NO$_X$ levels. The last seems improbable given the relatively good simulations of NO$_X$ mixing ratios at some sites. Investigation of the other possibilities is an area that needs future study, but is beyond the scope of this paper. It is important to note that the RACM17 chemistry successfully simulated O$_3$ events in the UB when observed NO$_X$ and speciated VOCs were input (Ahmadov et al.,

410    2015).

---

[3]methylperoxy radical, methyl hydroperxide, formaldehyde, methanol, ethene, ethan, acetaldehyde, ethanol and its oxides, acetic acid, glyoxal, glycolaldehyde, ethylperoxy radical, ethyl hydroperoxide, acetylperoxy radical, peracetic acid, peroxy acetyl nitrate, propene, propane and its oxides, acetone, hydroxyacetone, methylglyoxal, organic nitrate, lumped alkenes (C>3), methyl ethyl ketone and its oxides, methyle vinyl ketone, methacrolein, methacryloyl peroxynitrate, peroxy radicals, lumped alkanes (C>3) and their oxides, isoprene, unsaturated hydroxyhydroperoxide, lumped unsaturate hydroxycarbonyl, unsaturated dicarbonyl, lumped isoprene nitrate, lumped aromatics an their oxides, and lumped monoterpenes and their oxides

[4]ethane, alkanes, alcohols, esters, alkynes, ethene, terminal alkenes, internal alkenes, butadiene and other anthropogenic diens, isoprene, alpha-pinene and other cyclic terpenes, delta-limonene and other cyclic diene-terpenes, toluene, xylene, cresol, formaldehyde, acetaldehyde, ketones, glyoxal, methlglyoxal and other alpha-carbonyl aldehydes, unsaturated dicarbonyls, methacrolein and unsaturated monoaldehydes, unsaturated dihydroxyl dicarbonyl, hydroxy ketone, organic nitrate, preoxyacetyl nitrate and higher saturated PANs, unsaturates PANs, methyl hydrogen peroxide, higher organic peroxides, peroxyacetic acid, formic acid, acetic acid and higher acids, methyl peroxy radicals, aromatic peroxy radicals, acetyl peroxy and its saturated and unsaturated radicals





The spatial variation in the formation and dissipation of $O_3$ and its precursors for the high $O_3$ event on Mar 4, 2017, is shown in Figures 9, 10, and 11 for $O_3$, $NO_X$, and VOCs, respectively, from the MOZ17 simulation, and similarly, Figures 12, 13, and 14 show the results from RACM17. In both simulations, the formation and build up of $O_3$ is seen around noon local time (Figure 9c and Figure 12c), In the late afternoon (at 16:00 local time) the $O_3$ concentration reaches its maximum 110

ppb in MOZ17 (Figure 9d) and 95 ppb in RACM17 (Figure 12d). The $0_3$ concentration in RACM17 dissipates rather quickly compared to MOZ17 demonstrating that there are subtle differences in the chemical mechanisms. For both simulations, the highest $O_3$ concentration is seen closer to the Big Piney, Boulder, Daniel South and Pinedale stations, though none of the stations are simulated to observe the highest concentrations. If compared closely with the well locations in Figure 1, the highest $O_3$ concentrations overlap the location of the wells. The simulations show a similar temporal trend in $O_3$ formation,

which can also be seen in Figure 6, although the highest concentrations differ by approximately 30 ppb. The $O_3$ mixing ratios at Juel Spring, Moxa Arch, and South Pass are comparatively lower. The wind speeds are also stronger ($> 5$ ms$^{-1}$) around these stations. Particularly, around South Pass, the wind speeds are around 15 ms$^{-1}$. With the lack of mountains surrounding these stations and comparatively higher wind speeds, pollutant concentrations can be easily diluted and dissipated.

To better understand the formation, accumulation, and dissipation of $O_3$ precursors, i.e., $NO_X$ and VOCs, the diurnal and

spatial variations are shown for both simulations. The simulations suggest that, as expected, most $NO_X$ sources are in the production region for oil and gas, though the Pinedale results show that the inventory is missing some anthropogenic sources of $NO_X$, especially residential wood burning. The high concentrations of $NO_X$ along the bottom of the figures are due to Interstate 80 and not oil and gas infrastructure. Both chemical mechanisms show a similar trend in $NO_X$ with the build up of $NO_X$ concentrations in the morning at 08:00 local time (Figures 10b and 13b) the higher concentrations at noon local time

(Figures 10c and 13c), a few hours before the higher concentrations of $O_3$ are simulated, and the lower pollutant concentration at 16:00 local time (Figures 10d and 13d) when the $O_3$ concentrations are the highest. It is important to note that the simulations capture the lower $NO_X$ concentrations reasonably well and the simulated $NO_X$ concentrations do not vary largely among the simulations using different chemical mechanisms. Similar to the diurnal $NO_X$ profile, the diurnal profile of VOCs from both simulations (Figures 11 and 14) also shows a similar trend in the distribution of VOCs in the basin, with higher VOC

concentrations occurring a few hours before the higher $O_3$ concentrations are simulated. Overall the simulations capture the diurnal variation of the $O_3$ and its precursors reasonably well, however, the simulated concentrations of the precursors are lower compared to the respective observations.

## 4 Conclusions

Over the past decade, there have been a number of elevated wintertime $O_3$ events in the UGRB, WY, with concentrations often

exceeding 70 ppb and occasionally exceeding the 8-hour NAAQS. Ozone events, though much less severe than previously, have continued despite significant efforts to reduce emissions from oil and gas production. This drives the need for a photochemical model to better understand what is happening. This study, the the best of the authors' knowledge, is the first to utilize the EPA-NEI2014v2 emissions inventory with a fully coupled meteorology and chemistry model (WRF-Chem) to simulate $O_3$





events in the UGRB. Additionally, this study compared the results of two different chemistry mechanisms (MOZART and
RACM), focusing on their ability to replicate the concentrations of $O_3$. Neither chemistry mechanism can reproduce these high
$O_3$ events without modifying the default surface albedo of the base model. Furthermore, the dry deposition of gas species in
RACM was modified to better represent slower losses to snow surfaces.

For our analysis, we focused on a several-day period in 2017 in which the $O_3$ concentrations exceeded 70 ppb repeatedly
(Mar 3 to Mar 7, 2017). The WRF-Chem simulations were compared with the observations from 7 weather and air quality
monitoring stations operated by WYDEQ and located in the basin.

The model meteorology was first validated using the vertical profile of observed temperature during two IOP periods (Feb 28
to Mar 2 and Mar 9 to Mar 12 2011). Although the simulated temperature is 2 to 4 °C warmer than the observed temperature, the
simulation captured the inversion layer near the surface. Furthermore, to validate the the model's ability to predict the surface
meteorology, 2-m temperature and wind speed from two WRF-Chem simulations (MOZ17 and RACM17) were compared
with the observations at 7 weather stations. The simulated 2-m temperature showed a good correlation with the observation
at all stations. The simulated periods of low wind speeds also showed good agreement with the observed calm winds, though
variability in the exact magnitude of the low winds results in relatively poor correlation coefficients.

To study the model's ability to replicate high $O_3$ events, we analyzed concentrations of $O_3$ and its precursors ($NO_X$ and
VOCs). The simulations captured the high $O_3$ concentrations on Mar 4 reasonably well at most of the stations. The MOZ17
simulation better matched the observed $O_3$ concentrations, whereas the RACM17 simulation underpredicted the high $O_3$ con-
centrations. While the simulations captured essential trends in $NO_X$ and NMHC, they underestimated the concentrations,
especially for NMHC. The lower concentrations of $NO_X$ were simulated well, but the higher concentrations were underpre-
dicted, presumably because of missing sources in the inventory. Both chemistry mechanisms underpredicted NMHC by a factor
of 6, suggesting that the inventory poorly quantifies these emissions. Spatial plots of $O_3$ and its precursors show the predicted
spatial extent of $O_3$ formation and that the models suggest the monitoring sites are close to, but not at, the location of maximum
$O_3$.

Overall, the WRF-Chem simulations (MOZ17 and RACM17) were able to simulate $O_3$ formation during this event, which is
somewhat surprising given that the models had NMHC levels roughly six times lower than indicated by theobservations. This
suggests emissions in the NEI2014v2 dataset are too low and perhaps sources are missing in the emission inventory. Further
study of the sensitivity of the simulations to $NO_X$ mixing ratios and NMHC mixing ratios and reactivity are needed. Because
the RACM chemistry has previously been shown to perform reasonably well at simulating $O_3$ events in the UB (Ahmadov
et al., 2015), this study presents the possibility that $O_3$ might be able to be formed in the UGRB at significantly lower NMHC
levels than are currently observed, though further study is needed to confirm this.

*Code and data availability.* The WRF and WRF-Chem models are freely available online (https://github.com/wrf-model/WRF). The emis-
sion preprocessing tools and NEI emission data can be found at https://www2.acom.ucar.edu/wrf-chem/wrf-chem-tools-community. The
WYDEQ data can be obtained from https://www.wyvisnet.com.



*Author contributions.*  SG, ZL, and SM designed the study and conducted the model simulations, analysis, and comparison with observations.
TT and SR assisted with the model configuration and setup.

*Competing interests.*  The authors declare that they have no conflict of interest.


*Acknowledgements.*  We acknowledge Alison Eyth and Barron H. Henderson at the U.S. Environmental Protection Agency (EPA) for making
SMOKE outputs available and to Gabriele Pfister and Stacy Walters at the National Center for Atmospheric Research (NCAR) and Stu Mc-
Keen at the National Oceanic and Atmospheric Administration (NOAA) for developing and providing tools to integrate SMOKE emissions
into WRF-Chem.

We would like to acknowledge the use of computational resources (doi:10.5065/D6RX99HX) at the NCAR-Wyoming Supercomputing
Center provided by the National Science Foundation and the State of Wyoming, and supported by NCAR's Computational and Information
Systems Laboratory.

    The authors would like to acknowledge Dr. Gabriele Pfister from Atmospheric Chemistry Observations and Modeling Lab (ACOM),
National Center for Atmospheric Research (NCAR) and Dr. Ravan Ahmadov from National Oceanic and Atmospheric Administration
(NOAA) for their guidance and advice.



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

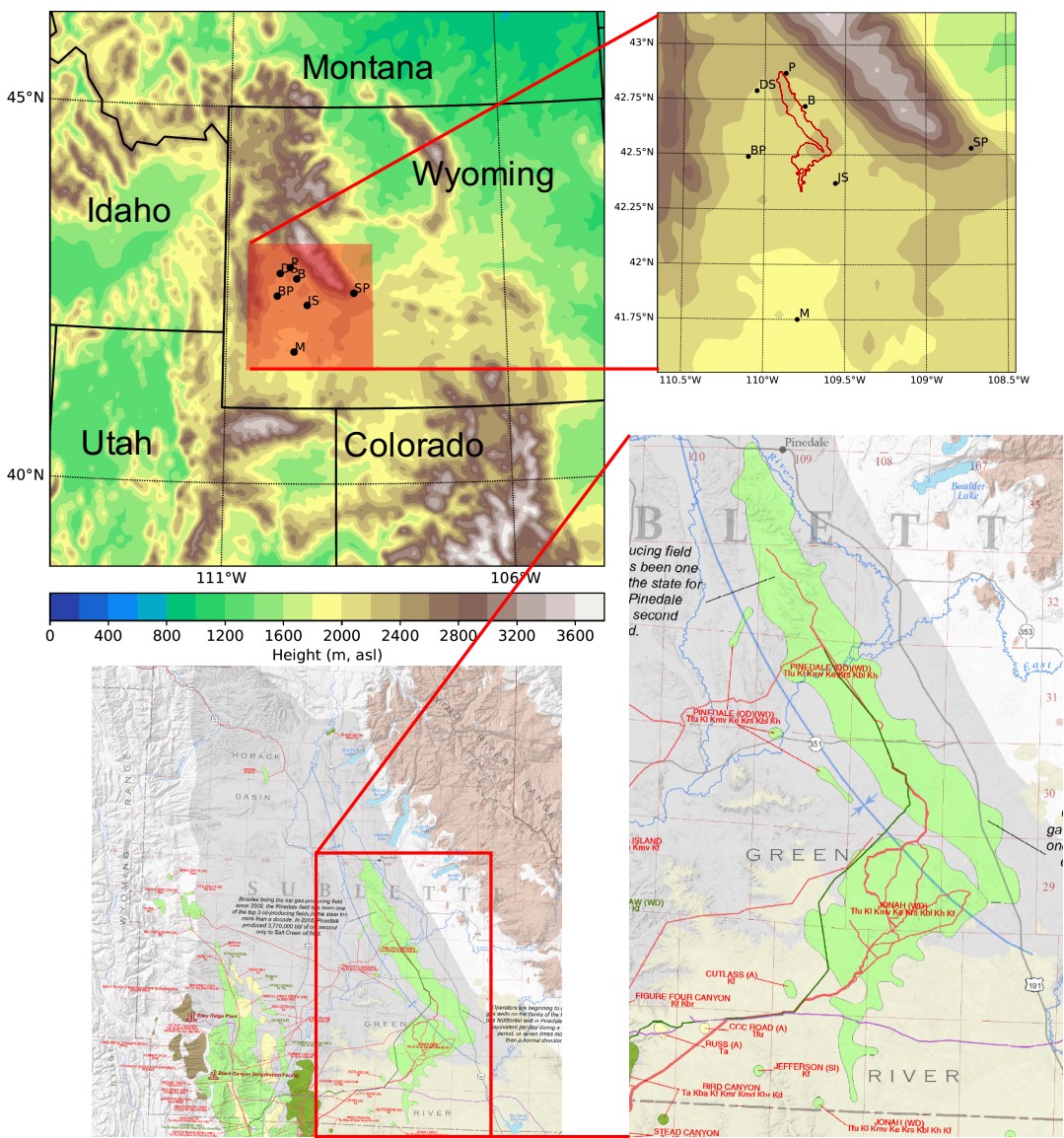

**Figure 1.** WRF domain (4 km x 4 km grid spacing) with WRF-derived terrain height (upper panels), along with 7 weather and air quality monitoring stations in Upper Green River Basin (shown by the red box). The red outline on the top-right plot is the approximate location of the Pinedale and Jonah Anticline Fields derived from the WSGS data depicted in the lower panels. The exact locations of the oil and natural gas wells in UGRB are also shown for reference in the bottom panels. The oil and gas facility data depicted in the lower panels are from Toner et al. (2019), ©WSGS.







**Figure 2.** The vertical profile of $O_3$ (ppb, green) and temperature (°C, red) from ozonesondes launched in 2011 by WYDEQ compared to WRF-simulated temperature (°C, blue) for 4 days. Each row represents 3-4 ozonesondes launched in one day.





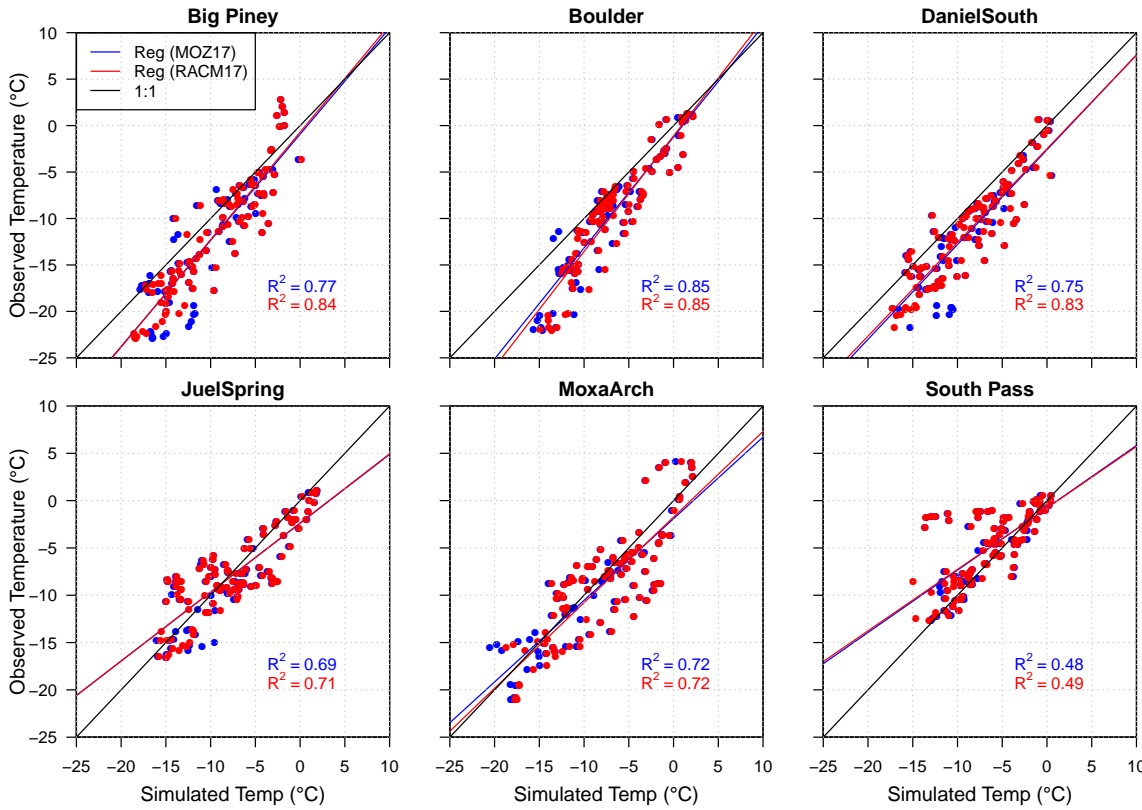

**Figure 3.** Correlation between recorded and simulated 2-m temperature at six monitoring stations. The data points and regressing line for MOZ17 are shown in blue and same for RACM17 are shown in red. The one-to-one lines are represented by black lines in each plot.





**Figure 4.** Similar to Figure 3 but for wind speed.





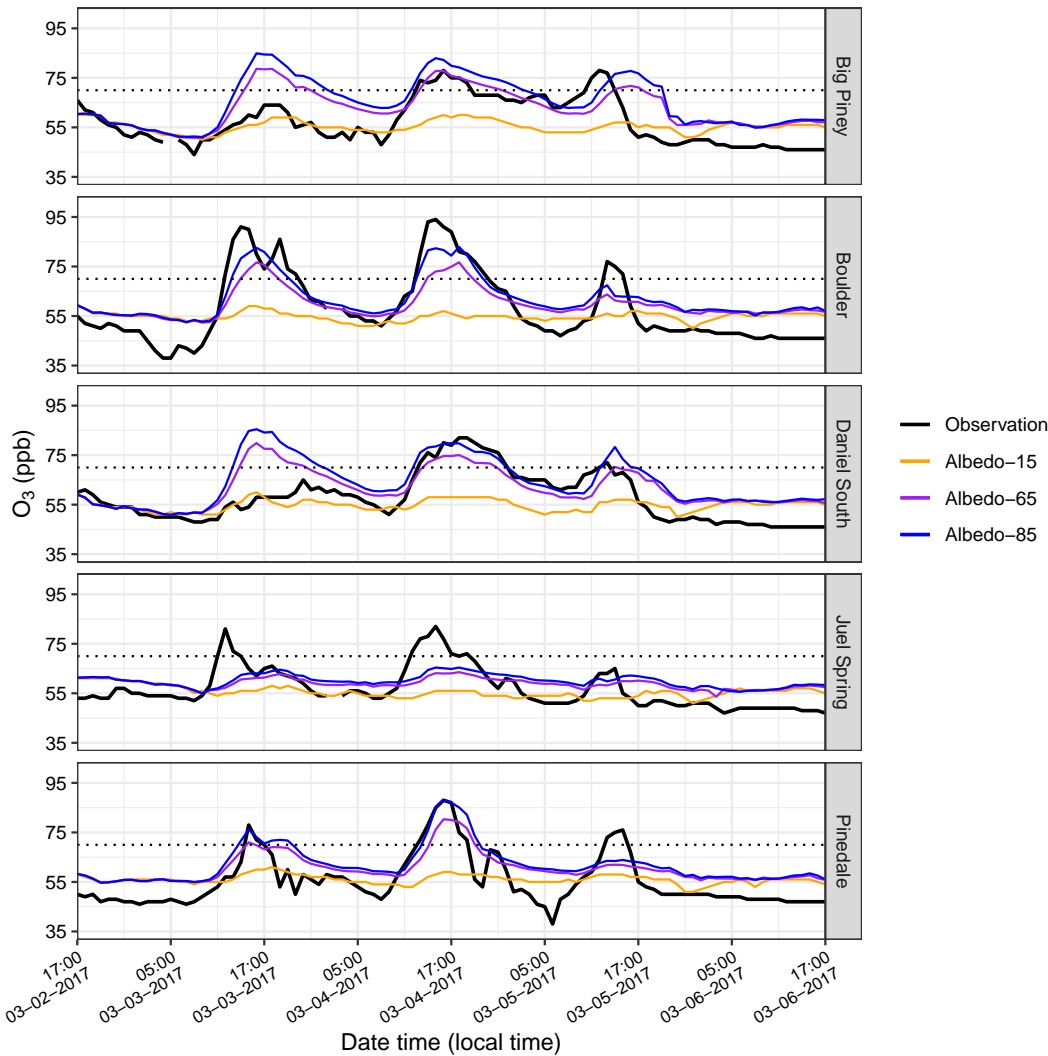

**Figure 5.** Albedo sensitivity for the WRF-Chem simulation at five monitoring stations. The observed O₃ concentrations at each station are shown in black lines, the orange lines represent the results from the default photolysis albedo of 0.15, and the purple and blue lines are the modified photolysis albedos of 0.65 and 0.85, respectively. The NAAQS 2015 standard is shown by the black dotted lines on each plot.





**Figure 6.** Time series of $O_3$ concentrations at 7 monitoring stations for the time period of Mar 3 to Mar 7, 2017, along with the 8-hour National Ambient Air Quality Standard, 2015 (dotted black lines). Similar to Figure 3, MOZ17 is represented by blue lines and RACM17 by red lines. The figure also shows the sensitivity of dry deposition in the RACM mechanism, with purple lines representing the simulation with RACM dry deposition turned on.







**Figure 7.** Similar to Figure 6 but for $NO_X$ concentrations (note the different y-scale for each station).





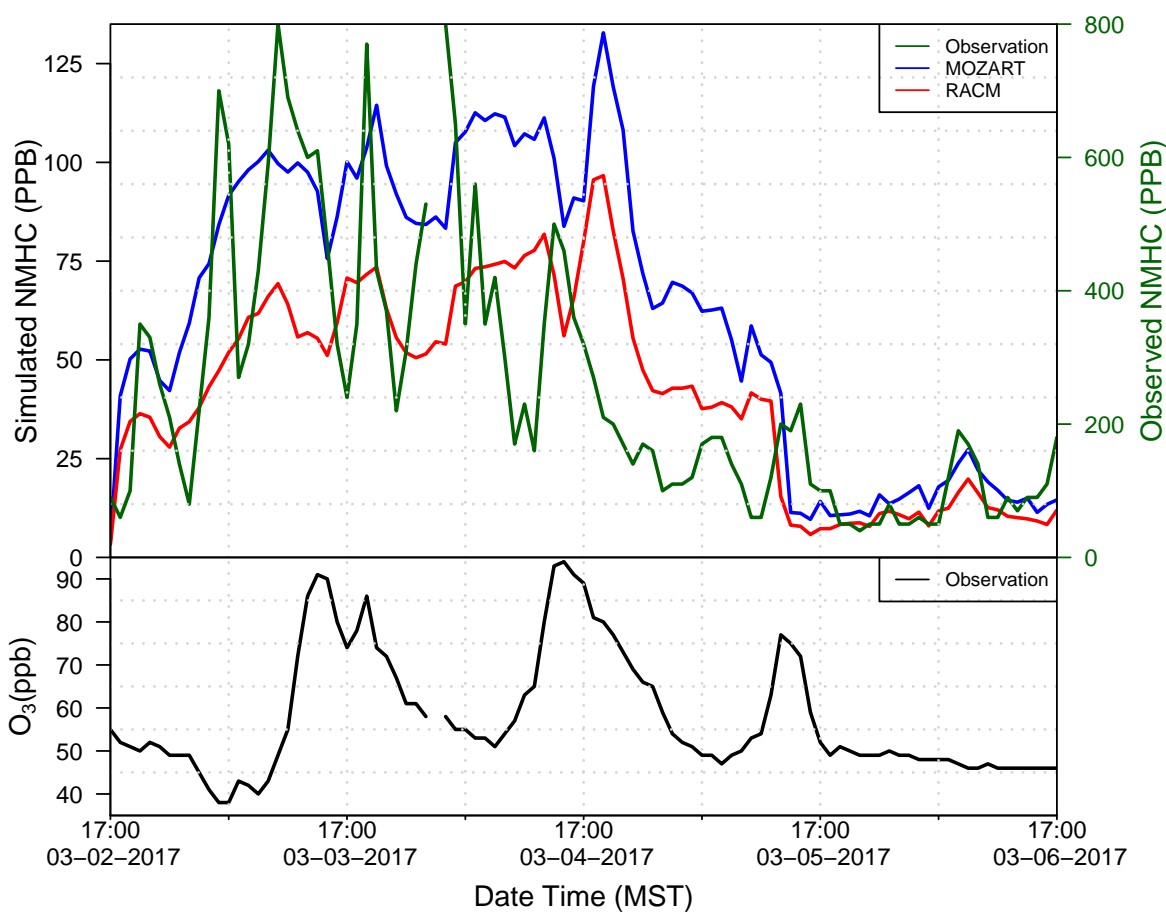

**Figure 8.** Time series of NMHC (top) and O₃ (bottom) at the Boulder site.





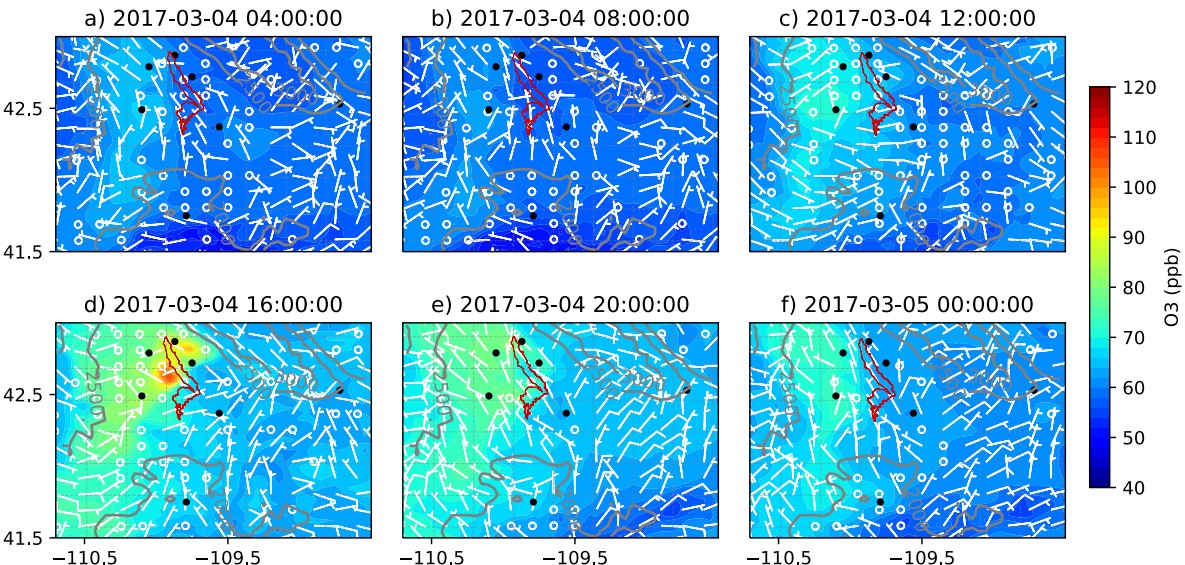

**Figure 9.** The formation and dissipation of O₃ concentrations over the basin using MOZART chemistry for the $O_3$ event on Mar 04, 2017, starting at 04:00 and ending at 24:00, with an interval of four hours in two consecutive figures. All times in the figure are in local time (UTC - 7 hours). The black dots are the location of the 7 WYDEQ stations, and the red outline is an approximate location of the PAJF development.





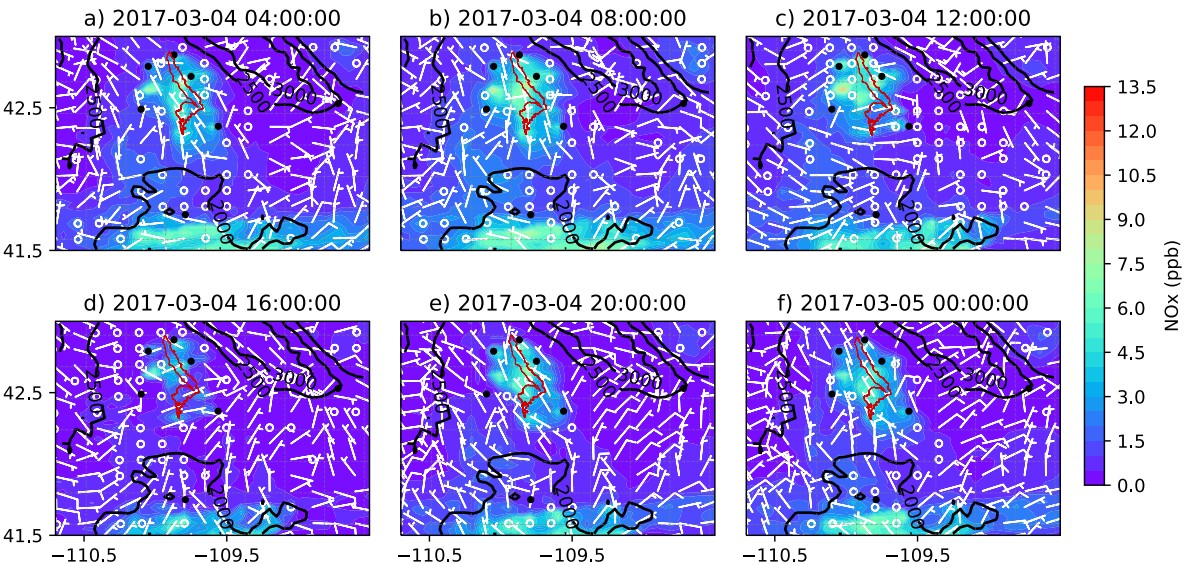

**Figure 10.** Similar to Figure 9 but for NO$_X$ concentrations.





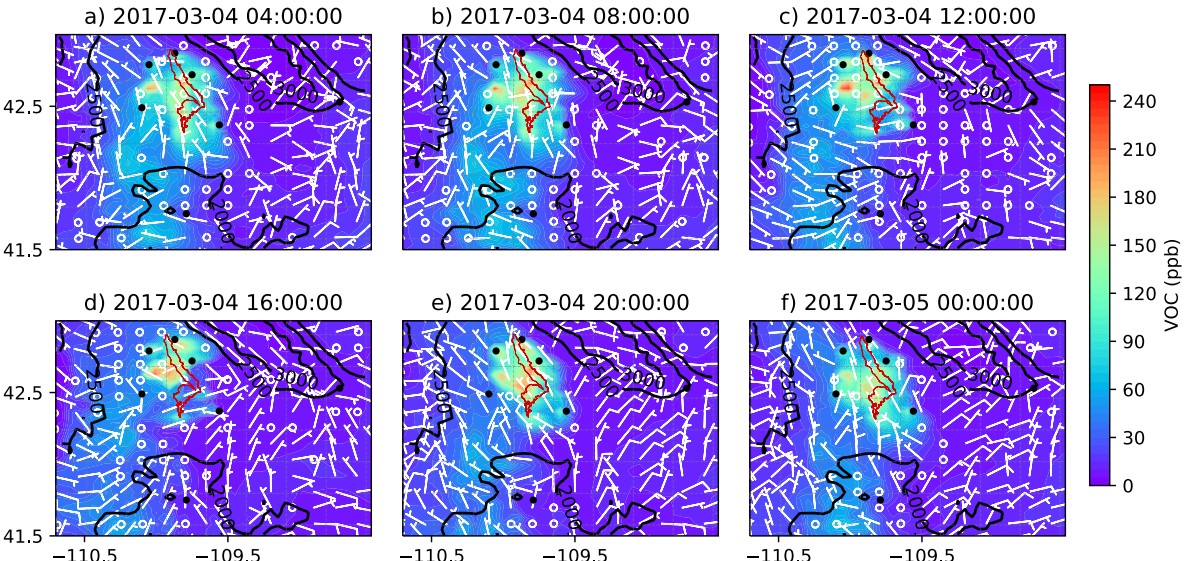

**Figure 11.** Similar to Figure 9 but for the concentrations of VOCs.





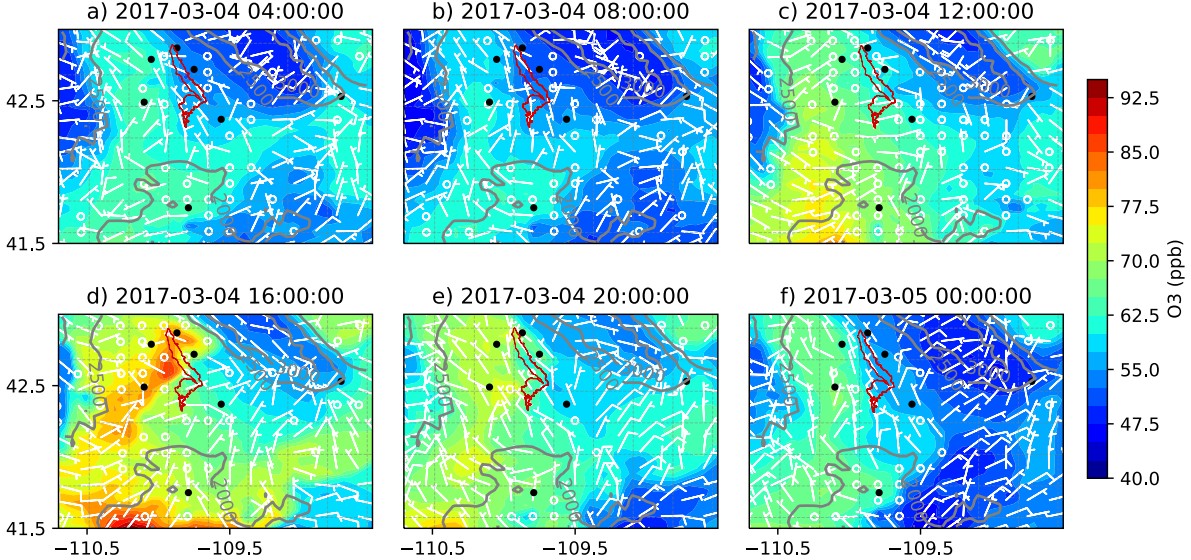

**Figure 12.** The simulated O$_3$ concentrations over the UGRB using RACM chemistry for the O$_3$ event on Mar 04, 2017.





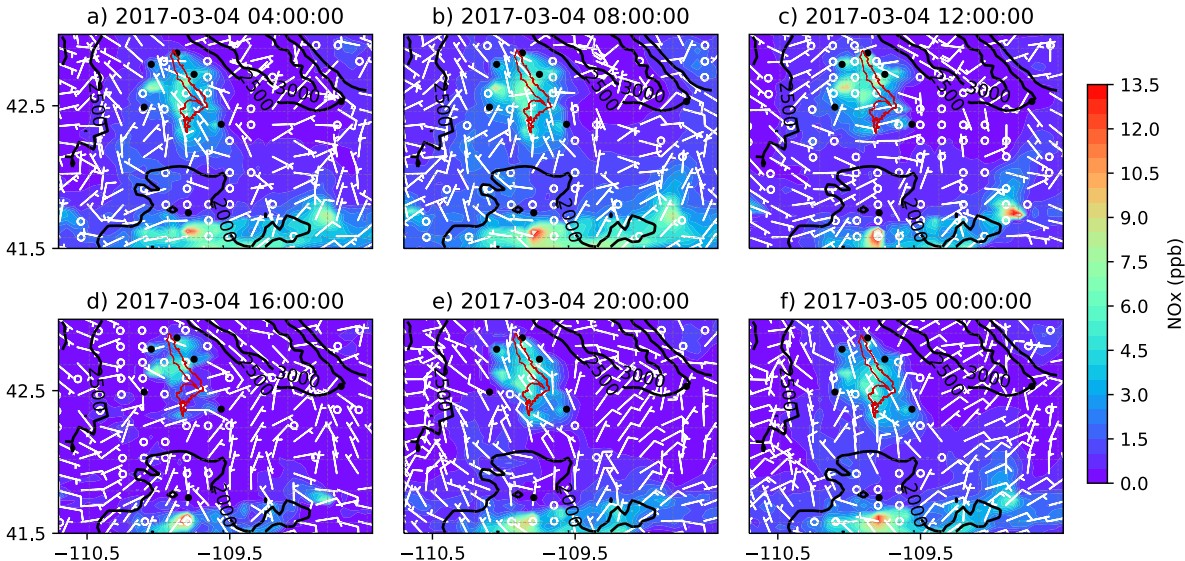

**Figure 13.** Similar to Figure 12 but for NO$_X$ concentrations.





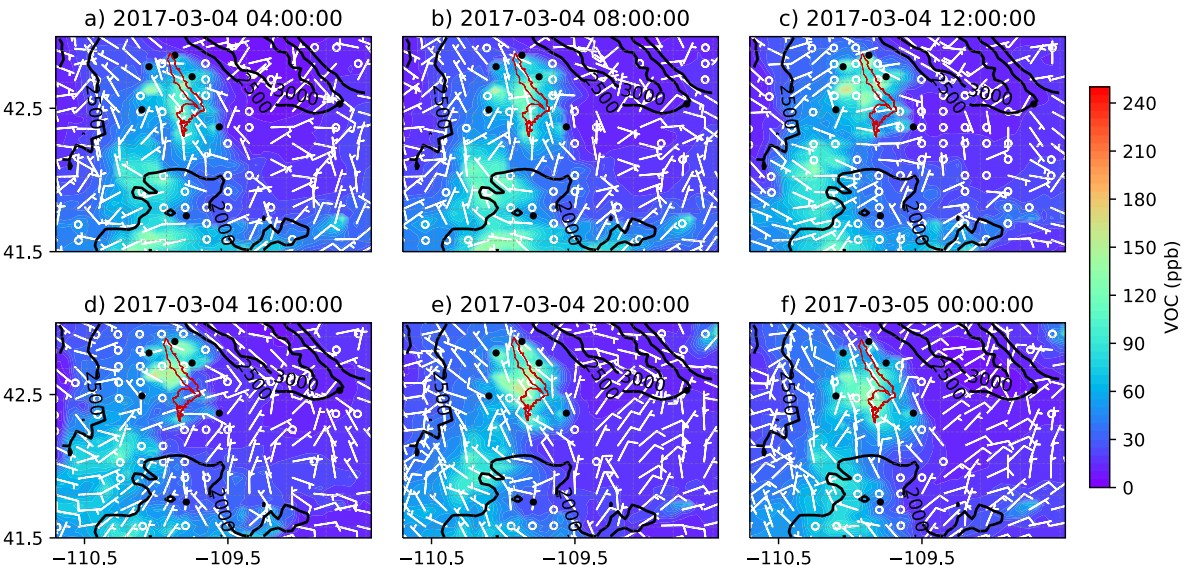

**Figure 14.** Similar to the Figure 12 but for VOC concentrations.





## List of Tables

**Table 1.** The coordinates and elevations of each weather and monitoring station in the UGRB. (*Source: www.wyvisnet.com*)

| Station | Latitude (°N) | Longitude (°W) | Elevation (ft) |
|---|---|---|---|
| Big Piney | 42.49 | 110.10 | 6,850 |
| Boulder | 42.72 | 109.75 | 7,110 |
| Daniel South | 42.79 | 110.05 | 7,129 |
| Juel Spring | 42.37 | 109.56 | 7,037 |
| Moxa Arch | 41.75 | 109.79 | 6,450 |
| Pinedale | 42.87 | 109.87 | 7,188 |
| South Pass | 42.53 | 108.72 | 8,287 |





**Table 2.** Model configuration for the base WRF Simulation

|  | Details |
| --- | --- |
| Boundary Conditions | NARR |
| Domain Size | 800 km x 800 km x 24 km |
| Time step | 12 |
| Horizontal Grid Spacing | 4 km (200 points x 200 points) |
| Vertical Levels | 60 (stretched) |
| Microphysics Scheme | Morrison double-moment scheme (Morrison et al., 2005) |
| Boundary Layer Scheme | MYJ (Janjić, 1994) |
| Radiation Scheme (LW and SW) | RRTMG (Iacono et al., 2008) |
| Land Surface Scheme | Noah-MP (Yang et al., 2011) |





**Table 3.** Temperature Bias (in °C) for the MOZ17 and RACM17 simulations.

|  | MOZ17 (°C) | RACM17 (°C) |
|---|---|---|
| Big Piney | 2.32 | 2.24 |
| Boulder | 2.60 | 2.79 |
| Daniel South | 2.77 | 2.56 |
| Juel Spring | 0.28 | 0.25 |
| Moxa Arch | 0.79 | 0.98 |
| South Pass | -1.43 | -1.50 |





**Table 4.** The percentage of the data points that are less than or equal to the given threshold (in m s$^{-1}$) when the observed wind speed is also less than or equal to the same threshold.

| Stations | MOZART | | | RACM | | |
|---|---|---|---|---|---|---|
| | <=3.0 | <=4.0 | <=5.0 | <=3.0 | <=4.0 | <=5.0 |
| Big Piney | 91.89 | 92.50 | 87.64 | 90.67 | 87.06 | 83.87 |
| Boulder | 89.39 | 83.54 | 80.49 | 92.19 | 86.84 | 74.16 |
| Daniel South | 85.29 | 94.74 | 83.91 | 75.00 | 92.31 | 82.95 |
| Juel Spring | 62.07 | 80.56 | 88.46 | 66.67 | 85.29 | 89.61 |
| Moxa Arch | 80.00 | 80.77 | 76.14 | 82.35 | 81.82 | 81.71 |
| Pinedale | 98.46 | 97.40 | 97.70 | 90.14 | 93.75 | 96.59 |
| South Pass | 38.46 | 50.00 | 67.35 | 40.00 | 46.15 | 61.11 |



**Appendix A: Data and methods**

**A1 Comparison with Mobile Laboratory Data**

Methane ($CH_4$) data from Picarro Cavity Ringdown Spectrometer (CRDS; model G2204) on-board University of Wyoming mobile laboratory Robertson et al. (2020) were used to validate the $CH_4$ concentrations from the Wyoming Department of Environmental Quality (WYDEQ) Boulder station. The CRDS was modified by Picarro Inc. to sample at 2-Hz. The National Institute of Standards and Technology (NIST) traceable ($\pm$ 1%) $CH_4$ in an ultrapure air mixture with a $CH_4$ concentration of 585 2.576 ppm was used to calibrate the Picarro instrument Robertson et al. (2020).

Due to data availability, we compared the hourly $CH_4$ data from WYDEQ with the 1-s data from the UW mobile laboratory. The data were from 11:00 am to 8:00 pm local time; the time period when UW mobile laboratory was driving in and around the UGRB.





**Figure A1.** Time series comparison of CH$_4$ from the UW mobile laboratory (red) and WYDEQ Boulder site (blue) for Feb 20, 2020. The black vertical lines mark the times when the mobile laboratory was passing through the WRF grid box where the WYDEQ Boulder site is located.





## A2 Chemistry namelist options used for MOZART and RACM chemistry mechanism

```
&chem
 kemit                                    = 1,
 chem_opt                                 = 202,
 bioemdt                                  = 30,
 photdt                                   = 30,
 chemdt                                   = 0,
 io_style_emissions                       = 2,
 emiss_inpt_opt                           = 102,
 emiss_opt                                = 10,
 emiss_opt_vol                            = 0,
 emiss_ash_hgt                            = 20000.,
 chem_in_opt                              = 1,
 phot_opt                                 = 4,
 gas_drydep_opt                           = 1,
 aer_drydep_opt                           = 1,
 bio_emiss_opt                            = 1,
 ne_area                                  = 178,
 dust_opt                                 = 13,
 dmsemis_opt                              = 1,
 seas_opt                                 = 2,
 depo_fact                                = 0.25,
 gas_bc_opt                               = 1,
 gas_ic_opt                               = 1,
 aer_bc_opt                               = 1,
 aer_ic_opt                               = 1,
 gaschem_onoff                            = 1,
 aerchem_onoff                            = 1,
 wetscav_onoff                            = 1,
 cldchem_onoff                            = 1,
 vertmix_onoff                            = 1,
 chem_conv_tr                             = 0,
 conv_tr_wetscav                          = 0,
 conv_tr_aqchem                           = 0,
 biomass_burn_opt                         = 3,
 plumerisefire_frq                        = 30,
 have_bcs_chem                            = .true.,
 aer_ra_feedback                          = 1,
 aer_op_opt                               = 1,
 opt_pars_out                             = 0,
 diagnostic_chem                          = 0,
 aer_aerodynres_opt                       = 2,
 has_o3_exo_coldens = .true.
 /
```

**Figure A2.** Namelist for chemistry options used for the simulation using MOZART chemistry mechanism.





```
&chem
 chem_opt         = 107,
 chem_in_opt     = 1,
 gaschem_onoff = 1,
 aerchem_onoff = 0,
 vertmix_onoff = 1,
 chem_conv_tr  = 0,
 gas_drydep_opt = 0,
 aer_drydep_opt = 0,
 diagnostic_chem = 2,
 chemdt  = 0,
 bioemdt = 30,
 emiss_inpt_opt  = 1,
 emiss_opt        = 3,
 kemit            = 10,
 io_style_emissions = 2,
 aircraft_emiss_opt = 0,
 bio_emiss_opt   = 0,
 phot_opt         = 1,
 photdt           = 30,
 wetscav_onoff   = 0,
 cldchem_onoff   = 0,
 conv_tr_wetscav = 0,
 conv_tr_aqchem  = 0,
 seas_opt         = 0,
 dust_opt         = 0,
 dmsemis_opt      = 0,
 biomass_burn_opt  = 0,
 have_bcs_chem      = .false.,
 gas_bc_opt         = 1,
 gas_ic_opt         = 1,
 aer_bc_opt         = 1,
 aer_ic_opt         = 1,
 aer_ra_feedback    = 0,
 opt_pars_out       = 0,
 /
```

**Figure A3.** Namelist for chemistry options used for the simulation using RACM chemistry mechanism.





**Appendix B:  Supplemental Figures**

**Figure B1.** Time series of albedo sensitivity for $O_3$ concentrations at 7 air quality monitoring stations. The black lines show the observed $O_3$ at different stations, and the orange, red, purple, blue and green lines represent photolysis albedos of 0.55, 0.65, 0.75, 0.85, and 0.95, respectively.



**Figure B2.** Time series of observation of $PM_{2.5}$ and $NO_X$ at Pinedale.