# Peer review of "Simulations of winter ozone in the Upper Green River Basin, Wyoming, using WRF-Chem"

_Atmospheric Chemistry and Physics, 2022_

## Author Comment (AC1)

We greatly appreciate the comments and criticism of the 2 anonymous reviewers and Dr. Seth Lyman. These reviewers provided critical suggestions that have substantially improved the quality of our paper. The main criticisms from the reviewers focused on 1) disparities between options selected for the different simulations (e.g., dry deposition, photolysis, boundary conditions, etc.) and 2) a lack of understanding of the sensitivity to different VOC and $NO_X$ emission scenarios. We have addressed both of these in detail in the revised manuscript. Specifically, we have done the following:

1. ensured consistency between options when running MOZART and RACM (when possible, as some namelist options do not work with one or the other chemistry mechanism)

2. identified dry deposition as a primary difference between the MOZART and RACM simulations such that when it is turned off in both cases, nearly identical levels of $O_3$ are produced

3. leveraged WYDEQ VOC and $NO_X$ data to quantify model biases and then adjust the emissions by these biases, enabling us to understand the sensitivity of $O_3$ production in the UGRB to different VOC and $NO_X$ conditions.

In addition to these major revisions to the manuscript, we have explicitly addressed all of the other comments from the reviewers and corrected minor grammatical errors that were overlooked in the original manuscript. We again thank the reviewers for the insightful feedback and hope that our revisions meet their expectations.

1. **Reviewer Comment:** This paper looks at high ozone events that occur in wintertime in the Upper Green River Basin. High wintertime ozone concentrations have been discussed in previous studies and are attributed to emissions from oil and gas extraction in combination with temperature inversions and enhanced photolysis fluxes due to snow covered ground. This paper discusses to what extent a regional chemical transport model (WRF-Chem) is able to represent these conditions by conducting sensitivity simulations with two different chemical mechanisms as well as a simulation where dry deposition has been turned off. The authors find that despite a significant underestimate in the modeled VOC concentrations, either chemical mechanism was able to represent the enhanced ozone concentrations.

This paper provides a good overview of previous work and in general the approaches and results are well presented. What I see missing is, however, a more in-depth analysis of the model results and an attempt to shed light into why the model despite a significant low bias in VOCs simulates ozone concentrations relatively well. The paper could be strengthened significantly by including more information on the NOx and VOc sensitivities in the model (e.g. looking at HCHO/NO2 ratios, modeled chemical tendencies etc.) and how they vary between the model simulations and also vary temporally and spatially. The model could also be compared to HCHO/NO2 ratios at the Boulder site if speciated VOCs are available (from the data set description it is not clear what type of VOC measurements were collected). It further would be valuable to focus on individual VOC species and not just the total VOCs

since the reactivity of different VOCs and their role in ozone production varies widely. The modeled VOC bias might be driven by only a few VOCs that have abundant emissions but play little role in ozone production.

**Author Response:** We thank the reviewer for the suggestions. As suggested, an in-depth VOC and $NO_X$ sensitivity analysis has been conducted. In the revised manuscript, the sensitivity simulations have been described in detail in Section 2.7, and the results have been discussed in Section 4. We are appreciative of this suggestion as the added results have strengthened the paper substantially.

*Specific Comments:*

2. **Reviewer Comment:** Line 143: I would disagree in that a valuable model should be able to represent conditions under any emission scenarios and VOC:NOx levels

   **Author Response:** The statement has been reworded on line 145 to read "*It is most useful to simulate $O_3$ events from recent years (versus modeling events in 2011) because basin-wide emission estimates from the State DEQ have decreased significantly over the last decade with potential impacts on both ozone precursor concentrations and VOC:NOx ratios. Also, we do not have emissions for oil and gas from 2011.*"

3. **Reviewer Comment:** Section Model Setup: Table 2 lists only a few of the settings and Figures A2 and A3 will only be meaningful to readers who are very familiar with WRF-Chem. I suggest extending Table 2 and explicitly stating some of the main information there. Additional information is also needed on the model configuration, e.g. what was used as chemical boundary conditions, was the meteorology in the model constrained and if so how, ...

   **Author Response:** Table 2 has been updated and now includes information regarding the chemical boundary conditions, dry deposition of gas species, lateral boundary conditions, and photolysis option used in the WRF-Chem simulations. The Community Atmosphere Model with Chemistry (CAM-Chem) data were used to update the chemical initial and boundary conditions in the model simulations, and this information has also been included in the revised paper. Specifically, in Section 2.3, line 189, the following has been added: "*To account for the transport of chemical species into the model domain, data from the Community Atmosphere Model with Chemistry (CAM-CHEM; Emmons et al. (2020)) were used in the simulations.*" Moreover, we have also added the following on line 248: "*The initial and boundary conditions of the simulations were updated every 24 hours for each simulations using the CAM-Chem data.*"

4. **Reviewer Comment:** Some questions to A2 and A3: The RACM setup does not use biogenic and fire emissions and also have_bcs_chem is set to false? There are a number of other differences between the MOZART and RACM simulation, so this means that the seen differences are not just related to the chemical scheme. Please elaborate on this and also provide justification behind these settings.

**Author Response:** We thank the reviewer for pointing this out. The simulations in the paper have been updated. Now, the biogenic and fire emissions along with aerosol radiation feedbacks have been turned off and have_bcs_chem has been set to true in all the simulations used in the revised version of the paper.

5. **Reviewer Comment:** In addition, this is a fairly small domain and I wonder how do chemical boundary conditions influence the ozone concentrations in the Basin? How were the chemical initial and boundary conditions treated (related also to comment above)?

   **Author Response:** As mentioned in the response for a comment above (comment #3), the chemical initial and boundary conditions were updated every 24-hr using the CAM-CHEM dataset.

6. **Reviewer Comment:** Section 2.3: More detail on the measurement techniques and the accuracy of the measurements is needed.

   **Author Response:** We agree we have not provided details on the measurement techniques. As mentioned on line 192, we have used the data provided by the Wyoming Department of Environmental Quality (WYDEQ), which was readily available on their website (www.wyvisnet.com) at the time this research was started.

7. **Reviewer Comment:** Line 198: MOZ17 has not yet been defined

   **Author Response:** The authors thank the reviewer for pointing this out. In the revised manuscript, MOZ17 is not mentioned until Section 3.2 on line 407, which reads as follows "......MOZ_ddOff and RACM_ddOff simulations will be discussed in the following analyses and the simulations will be referred to as **MOZ17** and **RACM17** respectively."

8. **Reviewer Comment:** Line 209: Was a spin-up period considered and if so how long?

   **Author Response:** The model was run for 4 days, and the results are generally consistent across all days. The only caveat is the overestimation of $O_3$ on the first day, which may be related to the model spinup time. However, given the model simulations are longer that this, we are confident the results for the later days are robust.

9. **Reviewer Comment:** Line 310: Is the model able to represent the diurnal variability and day-to-day variations? Is there a significant difference between daytime and nighttime performance?

   **Author Response:** On line 352, a statement has been added that reads "*An analysis of the diurnal variability of winds showed good qualitative agreement between the model and observations in terms of the timing of increasing and decreasing wind speeds each day (Figure not shown).*" The time series for observed and modeled wind speed is shown in Figure 1 in this document, but is not shown in the manuscript.

10. **Reviewer Comment:** Line 321: I suggest replacing "accurately" with adequately given that the model has clear shortcomings in representing observed conditions

**Author Response:** Suggested change has been made on line 356, which now reads as follows: *"Given the aforementioned ability of the model to adequately simulate the key meteorological..."*

11. **Reviewer Comment:** Line 339: I suggest to also define an acronym for the RACM simulation without dry deposition, e.g. RACM17_nodep to be consistent with the naming of the other simulations

    **Author Response:** Several acronyms are introduced in the paragraph starting on line 389. The MOZART and RACM simulations including dry deposition of gas species are defined as MOZ_ddOn and RACM_ddOn, and those that do not include dry deposition as MOZ_ddOff and RACM_ddOff, respectively.

12. **Reviewer Comment:** Line 374: I suggest a phrasing of "... do not show a strong sensitivity ..."

    **Author Response:** Reworded as suggested on line 417 as follows: *"The simulated concentrations of $NO_X$ **seems less sensitive** to the different chemical mechanisms,....."*

13. **Reviewer Comment:** Line 376: I would say that despite missing data there seems to be a clear overprediction in modeled NOx. Have the authors looked at whether the type of mapping the model 4km data to the site location could explain some of the differences between measured and modeled concentrations?

    **Author Response:** The statement regarding the overestimation of $NO_X$ at Daniel South was misleading. Both simulations predict same $NO_X$ concentration at the station. However, due to missing data points, it is not a good comparison. Hence, the statement regarding the overestimation of the $NO_X$ concentration at Daniel South has been reworded on line 419 as follows: *"The $NO_X$ mixing ratios are underestimated by both simulations even during the high $O_3$ events. Although the simulated $NO_X$ concentrations at Daniel South is higher compared to other stations, the observed data are missing."*

14. **Reviewer Comment:** Line 377: I suggest changing "removed" to "further away"

    **Author Response:** Suggested rewording has been done on line 421, which now reads as follows *".... emphasizing that this station is further from the oil and gas production region"*

15. **Reviewer Comment:** Line 379: Is this a boundary layer issue or an issue in the diurnal cycle of the emissions or is there any other reason?

    **Author Response:** The misleading statement has been reworded on line 419 as follows: *"The $NO_X$ mixing ratios are underestimated by both simulations even during the high $O_3$ events. Although the simulated $NO_X$ concentrations at Daniel South is higher compared to other stations, the observed data are missing. The observed and simulated $NO_X$ concentrations at South Pass are low and show little variability, emphasizing that this station is further from the oil and gas production region. Overall, the simulations underestimate*

*the observed $NO_X$ concentrations to varying degrees depending on the location and do not capture the diurnal cycle well.*"

16. **Reviewer Comment:** Line 381ff: Have you looked whether model grids surrounding the Boulder site have higher NMHC mixing ratios? More information is also needed on how the intercomparison was done and how it was ensured that the modeled total NMHC indeed reflect the same type of information as the measured NMHC (i.e. that it really is an apple to apple comparison)

    **Author Response:** We have looked at the model grid points surrounding the Boulder station and the NMHC mixing ratios are higher but not as high as the observations. In the revised manuscript, we compare the values of speciated VOCs with those from the model and adjust the emissions based on the factors calculated using these values. The detailed process of the calculation is discussed in Section 2.7, line 270.

17. **Reviewer Comment:** Line 388: What is the statement about NMHC removal based on?

    **Author Response:** Both MOZART and RACM simulations have similar NMHC mixing ratios. In order to understand how these chemical mechanisms lead to similar $O_3$ levels even with low mixing ratios of precursors, we performed VOC and $NO_x$ sensitivity simulations. As mentioned in previous comment #16, the detailed process is discussed in Section 2.7, and the analysis is presented in Section 4 . The major finding from this sensitivity study is mentioned on line 498, which states the following: "*These model runs strongly suggest that ozone formation in the basin is predominantly limited by the $NO_X$ available rather than being controlled mainly by VOC concentrations.*

18. **Reviewer Comment:** Figures 9-14: It is really hard to see any details in the NOx and VOC graphs, maybe a different color range could help? The paper also first discusses the ozone plots from Figure 12 and then looks into the NOx graphs. You might want to consider swapping the order of the Figures.

    **Author Response:** The color scheme and color range for Figures 9-14 (spatial plots for $O_3$, $NO_X$ and VOC for both MOZART and RACM simualtions) have been updated.

**References**

Emmons, L. K., Schwantes, R. H., Orlando, J. J., Tyndall, G., Kinnison, D., Lamarque, J.-F., Marsh, D., Mills, M. J., Tilmes, S., Bardeen, C., Buchholz, R. R., Conley, A., Gettelman, A., Garcia, R., Simpson, I., Blake, D. R., Meinardi, S., and Pétron, G.: The Chemistry Mechanism in the Community Earth System Model Version 2 (CESM2), Journal of Advances in Modeling Earth Systems, 12, e2019MS001 882, https://doi.org/10.1029/2019MS001882, URL       https://agupubs.onlinelibrary.wiley.com/doi/abs/10.1029/2019MS001882, e2019MS001882 2019MS001882, 2020.

[Figure]

Figure 1: Time Series of observed and simulated Wind Speed (m/s) at seven monitoring stations. NOTE: this figure is not included in the manuscript

---

## Author Comment (AC2)

We appreciate the comment from Dr. Seth Lyman, which motivated us to take a deeper look into VOCs and $NO_X$ in the WRF-Chem simulations of the UGRB. Our response to his comment is provided below:

1. **Reviewer Comment:** On lines 383-384, does the Boulder station measure speciated NMHC, or just methane and total NMHC? If it measures only total NMHC, I wouldn't trust the magnitude of the measurement. TNMHC GCs tend to strongly overestimate the amount of NMHC in the air compared to speciated measurements of individual compounds, so a comparison with that measurement is not likely to be useful.

   **Author Response:** Thank you for the insightful comment. Based on this comment and those from the other reviewers, we have included a sensitivity study in the revised manuscript focused on VOCs and $NO_X$. We have used WYDEQ data to adjust emissions of VOCs and $NO_X$ and compared the $O_3$ production. The result is that the basin appears to be highly constrained to the availability of $NO_X$.

---

## Author Comment (AC3)

We greatly appreciate the comments and criticism of the 2 anonymous reviewers and Dr. Seth Lyman. These reviewers provided critical suggestions that have substantially improved the quality of our paper. The main criticisms from the reviewers focused on 1) disparities between options selected for the different simulations (e.g., dry deposition, photolysis, boundary conditions, etc.) and 2) a lack of understanding of the sensitivity to different VOC and $NO_X$ emission scenarios. We have addressed both of these in detail in the revised manuscript. Specifically, we have done the following:

1. ensured consistency between options when running MOZART and RACM (when possible, as some namelist options do not work with one or the other chemistry mechanism)

2. identified dry deposition as a primary difference between the MOZART and RACM simulations such that when it is turned off in both cases, nearly identical levels of $O_3$ are produced

3. leveraged WYDEQ VOC and $NO_X$ data to quantify model biases and then adjust the emissions by these biases, enabling us to understand the sensitivity of $O_3$ production in the UGRB to different VOC and $NO_X$ conditions.

In addition to these major revisions to the manuscript, we have explicitly addressed all of the other comments from the reviewers and corrected minor grammatical errors that were overlooked in the original manuscript. We again thank the reviewers for the insightful feedback and hope that our revisions meet their expectations.

1. **Reviewer Comment:** The high wintertime O3 pollution in the Upper Green River Basin (UGRB), Wyoming is simulated in the study. During some years in winter months high O3 pollution in oil and gas producing basins of Utah and Wyoming have been observed. Numerous field campaigns and modeling studies have been conducted to understand the emissions and processes causing these high O3 pollution events. It is important for the air quality models to accurately simulate the wintertime O3 in UGRB, which could also help to develop mitigation strategies in the future. Here the authors deploy the-state-of-the-art WRF-Chem model to simulate high O3 during March, 2017. There are several aspects of the study that could make an important contribution to the field. The authors also conduct rigorous evaluation of the meteorological simulations. However, there are some shortcomings of the study that need to be addressed.

   **Author Response:** The authors thank the anonymous reviewer for the feedback provided.

   *Major Comments:*

2. **Reviewer Comment:** The authors emphasize the importance of using the existing anthropogenic emission inventories to model the high winter O3 in UGRB, and claim that this is the main strength of this study. While it's important to use the bottom-up emission inventories, the scientific community should not limit itself using the bottom-up inventories only. As Ahmadov et al. 2015 showed the EPA NEI inventory can grossly over/under-estimate

the NOx/VOC emissions from an oil and gas producing region (Uintah Basin). Therefore, in my opinion it's an underestimation of the importance of the study by focusing on the use of the emission inventory.

**Author Response:** We thank the reviewer for pointing out the findings from the Ahmadov et al. (2015) study in the Uinta Basin (UB), UT. We agree that the scientific community should not limit themselves to the emission inventories only, but we would like to point out that the studies using 3-D photochemical models (Ahmadov et al., 2015; Matichuk et al., 2017) have focused on the high $O_3$ events occurring in the UB not in the Upper Green River Basin (UGRB), WY. Also, as noted by several studies cited in the paper (Schnell et al., 2009; Oltmans et al., 2014; Rappenglück et al., 2014; Field et al., 2015; Lyman and Tran, 2015), the formation of $O_3$ in basins with operating natural gas and oil extraction facilities depend on the topography, meteorology, and chemical processes specific to the basin. These studies have pointed out that high $O_3$ events occur during different months in different basin and also vary from year to year in the same basin. Thus, the results of one basin cannot be applied to other basins without validation and analysis. In addition, the 3-D photochemical models used to study high $O_3$ events in the UB used a previous version of National Emission Inventory (NEI 2011) data compared to our study (NEI 2014v2). We would like to emphasize the use of the NEI 2014v2 dataset as these data include emissions from natural gas and oil production fields, which are not included in prior inventories. Thus, for these reasons, we would like to put an emphasis on the importance of using the existing athropogenic emission inventories to study a high $O_3$ event in the UGRB, WY.

3. **Reviewer Comment:** Introduction: The statement about the shortfalls of other studies is somewhat misleading. Do the authors refer to the box modeling studies conducted in the past? The box models are designed to use measured concentrations of the chemical species, not emission inventories. As for the 3D air quality models Ahmadov et. al. (2015) demonstrated that the emission inventories can have huge uncertainties. Moreover, as I discuss below this study doesn't prove that the NEI-2014 inventory accurately represents the emissions for the UGRB during March, 2017.

**Author Response:** We do refer to previous studies that use box models, as mentioned on line 51. Further, on line 94, the findings of these box models have been discussed.

4. **Reviewer Comment:** Here two different gas chemistry schemes are used, MOZART and RACM. As the WRF-Chem namelists provided in SI show the MOZART simulation included aerosols and their feedback on radiation. However, in the RACM simulations the authors turned off aerosols. In the paper differences in the meteorological simulations between these two model cases are presented and attributed to the aerosol feedback, though simulated aerosol fields aren't shown. I assume the aerosol concentrations in UGRB were relatively low.

**Author Response:** In the revised manuscript, we have updated all simulations used in this study. The major changes in the simulations include the use of the same namelist options

for both MOZART and RACM. Hence, in the revised MOZART simulation, aerosols and their feedback on radiation are not included. Despite this, there exists some differences in the meteorological fields, which is not due to the inclusion of aerosols and their feedbacks.

5. **Reviewer Comment:** The two gas chemistry mechanisms also use different photolysis schemes (phot_opt). Such difference makes it hard to compare the results of these two model cases.

   **Author Response:** A discussion regarding the use of different photolysis options has been provided in the revised paper, line 221: "*Despite all the same namelist options used in these models, the simulations with MOZART use photolysis option 4, which is the updated TUV photolysis option that was setup to work with only few chemistry mechanism schemes in WRF-Chem v3.9.1. While the RACM simulations use photolysis option 1, which is the Madronich photolysis scheme. With the current setup for photolysis option 4 in the WRF-Chem v3.9.1 it does not work with RACM chemistry mechanism. This study uses photolysis option 4 for MOZART simulation as it produces higher $O_3$ compared to when photolysis option 1 was used (Figure not shown).*"

   Furthermore, in Section 3.2, line 381, the difference in the results for the MOZART and RACM simulations owing to the use of different photolysis options has been discussed as follows: "*It is important to note that one difference between the MOZART and RACM simulations used in this study is the photolysis option (phot_opt = 4 for MOZART and phot_opt = 1 for RACM), which could affect $O_3$ production. As RACM is not coupled to phot_opt = 4, an addition sensitivity simulation was performed using option 1 with MOZART, which led to less $O_3$ compared with using option 4, albeit with better agreement with the observations. As such, we elected to use phot_opt = 4 for subsequent simulations but note that some of the difference between RACM and MOZART may be attributed to the photolysis scheme used with the former leading to less $O_3$ production.*"

6. **Reviewer Comment:** Here the model simulations are presented for 5 days only. This is quite short. I suggest extending the model simulations to evaluate the model's capability in simulating ozone and other chemical species other days in March, 2017. Even if O3 levels were low those days it's imporant to check the model's ability to simulate O3 and other species in different meteorological conditions by using the same model configuration and emission dataset.

   **Author Response:** We agree that extending the model simulation could provide more information on the model's capability at simulating $O_3$ and other chemical species. However, extending the simulation period is out of scope for this study as the model is computationally expensive and requires longer hours to run the simulations. Hence, a short period with one of the high $O_3$ events of the season was chosen for this study and the simulations were carried out in the finest resolution (in terms of model stability) possible. We do note that in lieu of extending the model period, considerable effort has been given to studying the sensitivity of $O_3$ formation to VOC and $NO_X$ emissions, which has greatly improved the

impact of the study.

7. **Reviewer Comment:** 350: Ahmadov et al. (2015) found that the reduced dry deposition of ozone over snow covered ground is one of key processes leading to high wintertime ozone buildup. It seems that the model has this snow impact on dry deposition in the MOZART scheme, but not in the RACM scheme in the version of the model used here. This discussion of the dry deposition needs to be revised.

   **Author Response:** The discussion on the dry deposition of gas species in both chemistry mechanisms has been updated in Section 3.2, line 387: "*To better understand the chemistry mechanisms' sensitivity to dry deposition, we compare the diurnal variation of $O_3$ concentrations from MOZART and RACM with dry deposition turned on and off in both simulations at the 7 monitoring stations.*" The results of these sensitivity simulations have also been described in the revised manuscript.

8. **Reviewer Comment:** Although the model is able to simulate the high O3, the simulated VOC mixing ratios are a factor of six lower than the observed ones. The NOx simulations show underestimation too. This begs the question, does the model simulate high O3 for the right reasons? It'd be helpful to conduct sensitivity simulations by adjusting the NOx and VOC emissions to account for uncertainties in the NEI.

   **Author Response:** Thank you for the great comment. We have performed a sensitivity study on VOC and $NO_X$ emissions. The adjustment process for this portion of the study is discussed in Section 2.7, line 270.

9. **Reviewer Comment:** 360: This is missing in the community version of WRF-Chem.

   **Author Response:** The line referenced in the original manuscript does not mention specific components of the WRF-Chem model, as such we are unsure what the reviewer is referencing.

10. **Reviewer Comment:** For the mitigation strategies it's helpful to understand the NOx/VOC sensitivity of the O3 formation. I suggest conducting sensitivity simulations by adjusting the emissions to show how the simulated O3 will respond to the NOx and/or VOC emission adjustments in UGRB.

    **Author Response:** Yes! Again, this is a great comment, which motivated us to explore the VOC and $NO_X$ parameter space in more detail. Sensitivity simulations with increased VOC and $NO_X$ emissions constrained to observations were conducted, and the results of these simulations are described in detail on line 495 of the revised manuscript.

11. **Reviewer Comment:** The advantage of using a tightly coupled meteorology-chemistry model such as WRF-Chem isn't discussed here. As Ahmadov et al. showed this is essential to simulate the stagnation episodes and multi-day buildup of the pollutants in a basin.

    **Author Response:** In Section 2.2, line 169, the benefit of using WRF-Chem has been updated as follows: "*This is beneficial over models such as CAMx and CMAQ where the*

*meteorological and the atmospheric chemistry components are run separately. Ahmadov et al. (2015) also pointed out the benefit of WRF-Chem, which helped in the proper simulation of pollutant accumulation in shallow inversion layers."*

*Minor Comments:*

12. **Reviewer Comment:** The CMAQ modeling paper by Matichuk et al. (https://doi.org/10.1002/2017. isn't cited here.

    **Author Response:** We have somehow missed to cite this CMAQ modeling paper and we would like to thank the reviewer for suggesting this article. We have added the citation for Matichuk et al. (2017) on line 119, "*Matichuk et al. (2017) used the WRF and Community Multiscale Air Quality (CMAQ) models to study a 10-day high-ozone episode in 2013 in the UB. Similar to Ahmadov et al. (2015), they also used the NEI2011 emission dataset, but they found that the CMAQ model did not reproduce the observed $O_3$, $NO_X$, and VOC levels in the UB. Matichuk et al. (2017) identified a positive temperature bias and overestimation of the daytime planetary boundary layer height in the WRF simulations, which was hypothesized to be the reason for underestimation of $O_3$, $NO_X$, and VOCs from the CMAQ model.*"

13. **Reviewer Comment:** The evaluation of the meteorological simulations can be moved to SI.

    **Author Response:** The authors think that meteorological simulations be kept in the main manuscript and hence have not been moved to SI. The reason is that $O_3$ formation is highly sensitive to the meteorological conditions, and we feel that is is important to show the model's ability to simulate the conditions up front.

**References**

Ahmadov, R., McKeen, S., Trainer, M., Banta, R., Brewer, A., Brown, S., Edwards, P., De Gouw, J., Frost, G., Gilman, J., et al.: Understanding high wintertime ozone pollution events in an oil-and natural gas-producing region of the western US, Atmospheric Chemistry and Physics, 15, 411–429, 2015.

Field, R., Soltis, J., Pérez-Ballesta, P., Grandesso, E., and Montague, D.: Distributions of air pollutants associated with oil and natural gas development measured in the Upper Green River Basin of Wyoming, Elem Sci Anth, 3, 2015.

Lyman, S. and Tran, T.: Inversion structure and winter ozone distribution in the Uintah Basin, Utah, USA, Atmospheric Environment, 123, 156–165, 2015.

Matichuk, R., Tonnesen, G., Luecken, D., Gilliam, R., Napelenok, S. L., Baker, K. R., Schwede, D., Murphy, B., Helmig, D., Lyman, S. N., et al.: Evaluation of the community multiscale air quality model for simulating winter ozone formation in the Uinta Basin, Journal of Geophysical Research: Atmospheres, 122, 13–545, 2017.

Oltmans, S., Schnell, R., Johnson, B., Pétron, G., Mefford, T., and Neely III, R.: Anatomy of wintertime ozone associated with oil and natural gas extraction activity in Wyoming and Utah, Elementa: Science of the Anthropocene, 2, 2014.

Rappenglück, B., Ackermann, L., Alvarez, S., Golovko, J., Buhr, M., Field, R., Soltis, J., Montague, D. C., Hauze, B., Adamson, S., et al.: Strong wintertime ozone events in the Upper Green River basin, Wyoming, Atmospheric Chemistry and Physics, 14, 4909, 2014.

Schnell, R. C., Oltmans, S. J., Neely, R. R., Endres, M. S., Molenar, J. V., and White, A. B.: Rapid photochemical production of ozone at high concentrations in a rural site during winter, Nature Geoscience, 2, 120, 2009.

---

## Author Response (AR2)

**Reviewer 2**

The revision have improved the significance of the paper and I have only some remaining comments/questions before I see the paper as ready for publication.

**Author Response:** We thank the anonymous reviewer again for insightful and critical feedback on our manuscript. We originally envisioned this work being included in two papers: 1 demonstrating the model's ability to predict $O_3$ in the UGRB and 1 focused on VOC and $NO_X$ sensitivity. Based on the comments and suggestions, we have conducted the sensitivity analysis and included it in this study. We have fully explored both VOCs and $NO_X$ in this revision based on the comments and suggestions provided. Further, we have carefully reviewed the entire paper to 1) reduced unneeded repetition, 2) correct grammatical errors), 3) improve the flow of the text, and 4) address other minor text errors. We truly believe that this work will have a much larger impact on the community after the suggested additions/revisions.

1. **Reviewer Comment:** Why were the boundary conditions only updated every 24 hours if the global model output is available every 6 hours?

   **Author Response:** We apologize for the confusion. This was an error in the original manuscript. In fact, the boundary conditions were updated every 6 hours, and this has been revised accordingly on line 188: "*To account for the transport of chemical species into the model domain, 6-hourly data from the Community Atmosphere Model with Chemistry (CAM-CHEM; Emmons et al. (2020)) were used in the simulations.*"

2. **Reviewer Comment:** My question about the spin-up period was not really answered and at least a 1-2 day spinup should be considered in any study. At the least it should be mentioned that the first days might be affected by spin-up effects and that results for these days needs to be taken with caution.

   **Author Response:** We apologize for not fully addressing your original comment. In practice, 1-2 days of spinup is ideal. However, given the computational expense of the simulations and the growing number of sensitivity simulations, additional simulation days preceding the study period for spin-up were not considered. However, $O_3$ does not start ramping up until nearly 24 hours into the simulation, and the results of the peaks in $O_3$ are generally qualitatively similar between days providing some confidence that a longer spin-up period would not have changed the key outcomes of this work. To address this in the manuscript, a statement regarding the spin-up period has been added on line 197 under Section 2.4: "*A spin-up period was not explicitly considered in this study owing to the computational expense of each simulation and that $O_3$ generally does not start increasing until nearly 24 hours into the simulation; however, the results from the first day should still be viewed with caution.*"

3. **Reviewer Comment:** Evaluation of winds: I would not necessarily say that the model does a great job at simulating WS, specifically for the last day. The authors also did not provide information on wind direction. I suggest rephrasing "good qualitative agreement"

to something like "reasonable qualitative agreement" or similar. Also please comment on the evaluation for wind direction.

**Author Response:** Thank you for the comment. The comment on the last day is valid; however, on this day, $O_3$ levels are quite low and below the 70 ppb threshold commonly used to indicate a high $O_3$ event. The key is that the wind speeds are well simulated on the days of interest, i.e., the days with elevated $O_3$ contents. We feel that mentioning the small negative wind speed bias on the final day would be a distraction for the reader based on this.

We have changed the wording from "good qualitative agreement" to "*reasonable qualitative agreement*" in line 360.

Regarding the wind direction, the reviewer is correct that we did not provide information on validating the wind direction in the model simulations. The reason is the same as focusing the wind speed analysis on metrics such as the occurrence of wind speeds below a certain threshold over a direct correlation analysis. The key wind speed criterion is weak winds. We have shown in the paper that the model exhibits satisfactory performance with regard to low wind speeds. As such, we are far less concerned about the direction because if the wind speed is weak, the biases in the wind direction will have minimal impact on chemical transport. Moreover, at low wind speeds, wind direction observations can vary considerably and are less reliable. For these reasons, we feel that it is best to exclude a discussion of modeled versus observed wind directions in the manuscript. Future work is currently under way focused on detecting precursor leaks from oil and natural gas wells, and in this work we are focused on wind speed and detection because the work is intended to be generally applicable to all weather conditions, not just those conducive to high $O_3$ levels.

4. **Reviewer Comment:** Line 254: The photolysis option 4 should not be used because it gives higher ozone for this case study but because it is an updated parameterization based on recent advanced in the understanding of photolysis rates.

**Author Response:** This is a very good point, and our original wording certainly did not make this clear. The text in the manuscript has been updated on line 222 to read as follows: "*Where possible, the same namelist options were used for both models. However, regarding the photolysis option, the simulations with MOZART used photolysis option 4, which is the updated TUV photolysis option based on recent advanced in the understanding of photolysis rates that was configured to work with only a few chemistry mechanism schemes in WRF-Chem v3.9.1, while the RACM simulations used photolysis option 1, which is the Madronich photolysis scheme.*"

5. **Reviewer Comment:** Line 300ff: Models struggle getting the boundary layer correct and specifically so for early morning. This will impact the model evaluation. At this time of the day, I would assume the measurements to be strongly impacted by local effects because of limited mixing which I would not expect the model to represent due to coarser spatial

resolution. So if there is a nearby source at the Boulder station, the measured signals might be dominated by it rather than representative of the regional characteristics.

**Author Response:** The reviewer has a good point. We utilized the mobile lab data to confirm that the methane concentration at the site was similar to the methane in the larger grid. While methane is not a perfect proxy for VOCs, it generally tracks the smaller alkanes (ethane to pentane) fairly closely and the smaller alkanes represent the majority of the VOC mixing ratio. This has been explained more clearly on line 439 that reads as follows: *"The WYDEQ Boulder data were within 25% of the data collected by the mobile lab near the monitoring site (Fig. D1 in Appendix D). This observation indicates that the difference between the simulated and observed NMHC mixing ratios is not the result of anomalously high mixing ratios at the Boulder site, and thus the NMHC mixing ratio measured at the Boulder site is an accurate representation in the region."*

6. **Reviewer Comment:** Line 305: I am puzzled by the use of the word "emission factor". An emission factor typically represents the amount of a compound released, e.g. grams of a compound per biomass burned. What is shown here is simply the ratio of the observed and modeled value. I recommend using a different term, e.g. scaling factor or similar. I also caution that you are using ambient concentrations to correct the emissions without knowledge about the age of the airmasses and how representative they indeed are of fresh emissions.

**Author Response:** Yes, this is a very good point, and sorry for the confusion. All instances of "emission factor" have been revised to "emission adjustment factor" in the revised paper.

7. **Reviewer Comment:** Line 314: It needs to be explained here why the dry deposition was turned off (or at least referred to the later discussion on this)

**Author Response:** The use of dry deposition for the sensitivity studies was not mentioned until later in Section 4, where the results from the sensitivity simulations are analyzed and discussed. To address this, on line 392, the text has been reworded to read as follows: *"RACM does not adjust the dry deposition rate over snow-covered surfaces, hence the dry deposition is likely too high in RACM and could explain the underestimate of $O_3$. Thus, we turned off dry deposition to mimic the very slow deposition of gas-phase species over a snow-covered surfaces (i.e., RACM_ddOff )."*

8. **Reviewer Comment:** NOx Sensitivity Simulations: As stated, I am worried that the VOC measurements at the Boulder site might be significantly impacted by local effects and by adjusting the VOCs in the model for the entire area based on these values might not be truly representative of the region. Have the authors tested the $NO_X$ sensitivity also with the base emissions? At the very least they should comment on the changes in the $HCHO/NO_2$ ratios and potentially chemical regimes that arise from an increase in the VOC emissions.

**Author Response:** The reviewer brings up an important point about utilizing data from a single station to adjust VOC emissions. We have added Table B1, which shows the

observed-modeled VOC ratios for the other observations sites in the basin where canister data are collected. Further, in line 275 the following text has been added to the manuscript: "*Table B1 in Appendix B shows ratios of canister-observed speciated VOC mixing ratios to the simulated values. The Moxa Arch site is relatively far from the emission sources and is not representative of the main $O_3$ formation region. The other sites show variability, although all sites show significant underestimates of both VOCs and $NO_X$. The Boulder site has the largest underestimates of reactive BTEX species, while some species have larger underestimates at other sites. Given this comparison and the site-to-site variability in the model–observation comparison, VOCs measured at the Boulder site appear to be a reasonable, though aggressive, basis for adjusting the emissions in the model.*"

We also thank the reviewer suggesting that we focus also on $NO_X$ sensitivity alone. We have conducted two more simulations for each chemical mechanisms, which are described on line 501: "*To explore the sensitivity to $NO_X$ vs. VOC further, additional simulations were conducted where only $NO_X$ and only VOCs were adjusted from the baseline run (Fig. 16). The simulation where only $NO_X$ was adjusted results in moderate increases in the simulated $O_3$ mixing ratios at all sites and across the entire study period. When NO and $NO_2$ emissions were kept at their baseline levels and VOC emissions were adjusted, the simulations produce slight increases in $O_3$ at some sites, especially with the MOZART chemistry scheme, but at other sites the $O_3$ mixing ratios are not always elevated. Rather, in these cases, the timing of $O_3$ formation changes, with increases seen earlier in the day and sometimes even lower peaks in the modeled $O_3$ mixing ratio compared with the baseline run. These results are interesting and somewhat unusual, but further analysis was not pursued because by increasing VOCs while not adjusting $NO_X$, the VOC:$NO_X$ ratio for this model run is far outside what is actually observed in the basin and is thus considered highly unrealistic.*"

Further, we have also looked at the changes in the HCHO:$NO_2$ ratio. On line 512, the following discussion has been added: "*To further investigate this, the formaldehyde:$NO_2$ (HCHO:$NO_2$) ratio for all VOC and $NO_X$ sensitivity simulations is presented in Fig. **??**. This ratio has been used in previous studies as a proxy for VOC-limited and $NO_X$-limited conditions (Liu et al., 2021). Here, the ratio is well above 1 during the high $O_3$ events for all simulations, with the only decrease being observed for the simulations with only increased $NO_X$ emissions. These results further suggest that $O_3$ formation in the basin is strongly controlled by $NO_X$ availability (Liu et al., 2021)*"

9. **Reviewer Comment:** The wording for some of the added parts would need to be proofread and improved"

**Author Response:** We have carefully reviewed the entire manuscript to revise and improve the wording. Several paragraphs have been removed to eliminate redundancy, and a lot of the text has been revised to ensure grammatical correctness and conciseness.

**References**

Emmons, L. K., Schwantes, R. H., Orlando, J. J., Tyndall, G., Kinnison, D., Lamarque, J.-F., Marsh, D., Mills, M. J., Tilmes, S., Bardeen, C., Buchholz, R. R., Conley, A., Gettelman, A., Garcia, R., Simpson, I., Blake, D. R., Meinardi, S., and Pétron, G.: The Chemistry Mechanism in the Community Earth System Model Version 2 (CESM2), Journal of Advances in Modeling Earth Systems, 12, e2019MS001 882, https://doi.org/10.1029/2019MS001882, URL https://agupubs.onlinelibrary.wiley.com/doi/abs/10.1029/2019MS001882, e2019MS001882 2019MS001882, 2020.

Liu, J., Li, X., Tan, Z., Wang, W., Yang, Y., Zhu, Y., Yang, S., Song, M., Chen, S., Wang, H., et al.: Assessing the Ratios of Formaldehyde and Glyoxal to NO2 as Indicators of O3–NO x–VOC Sensitivity, Environmental Science & Technology, 55, 10 935–10 945, 2021.